# SCI-VERIFIER: SCIENTIFIC VERIFIER WITH THINKING

**Shenghe Zheng**[1,2*]**, Chenyu Huang**[3*]**, Fangchen Yu**[1,6]**, Junchi Yao**[1,7]**, Jingqi Ye**[1,8]**,
Tao Chen**[3]**, Yun Luo**[1]**, Ning Ding**[1,5]**, Lei Bai**[1]**, Ganqu Cui**[1]**, Peng Ye**[1,4] †

[1] Shanghai Artificial Intelligence Laboratory    [2] Harbin Institute of Technology
[3] Fudan University    [4] The Chinese University of Hong Kong    [5] Tsinghua University
[6] CUHK-Shenzhen    [7] University of Electronic Science and Technology of China
[8] University of Science and Technology of China

## ABSTRACT

As large language models (LLMs) are increasingly applied to scientific reasoning, the complexity of answer formats and the diversity of equivalent expressions make answer verification a critical yet challenging task. Existing verification studies in scientific domains suffer from two major limitations: (a) the absence of systematic evaluation standards and insufficient disciplinary coverage, which hinders their comprehensive assessment; and (b) heavy reliance on cumbersome rule design or prompt engineering, which reduces their effectiveness in complex reasoning scenarios or limits their cross-disciplinary generalization. To address these challenges, we propose solutions at both the data and model levels. On the data side, we construct **SCI-VerifyBench**, a cross-disciplinary benchmark covering mathematics, physics, biology, chemistry, and general scientific QA. The benchmark is built from real LLM responses and enhanced with domain-specific equivalence transformations that generate challenging and realistic data. Model-based and expert annotations ensure both quality and diversity, enabling rigorous evaluation of verification ability. On the model side, we emphasize the importance of reasoning for verification and introduce **SCI-Verifier**, a unified reasoning-augmented verifier for scientific domains. Through post-training, SCI-Verifier demonstrates strong logical reasoning and equivalence judgment capabilities while maintaining concise and stable outputs. Together, SCI-VerifyBench and SCI-Verifier provide a principled framework for scientific verification, offering both systematic evaluation and practical pathways to enhance the reliability and applicability of LLMs in scientific domains.

## 1 INTRODUCTION

As large language models (LLMs) become increasingly prevalent in scientific reasoning (Yang et al., 2025; Ren et al., 2025; Liu et al., 2024a; Bai et al., 2025), ensuring the reliability of their outputs has emerged as a critical challenge (Chang et al., 2024; Liu et al., 2025). Scientific reasoning often involves intricate multi-step processes and a wide range of equivalent answer formulations (Chen et al., 2025b), posing substantial difficulties for answer verification. The essence of verification lies in accurately determining whether an output of LLM is equivalent to the reference answer, a task that serves both as the foundation for evaluating capabilities of LLMs and as a key bottleneck to further advancement (Zhang et al., 2025a; Chen et al., 2025a; Zhang et al., 2025b).

Despite recent progress, verification research in scientific domains still faces two major challenges. First, high-quality and systematic benchmarks are lacking. Existing benchmarks cover only a narrow range of scientific disciplines and fail to account for discipline-specific equivalence forms (Liu et al., 2024b; Li et al., 2025; Yan et al., 2025; Chen et al., 2025a), making it difficult to comprehensively evaluate verification capabilities. Second, current methods are limited in complex reasoning scenarios and lack cross-disciplinary applicability. Rule-based approaches rely on manually crafted templates and heuristics, which are labor-intensive and insufficient for complexity (Hynek Kydlíček, 2024; Gao et al., 2021). More recently, general LLMs or specialized verifiers leverage LLM generalization to achieve promising results (Li et al., 2023; Liu et al., 2025; Zhang et al., 2025b; Luo et al., 2023;

---

*Equal Contribution.  Work done during the internship at Shanghai AI Lab.  Project: SCI-Verifier
†Corresponding Author

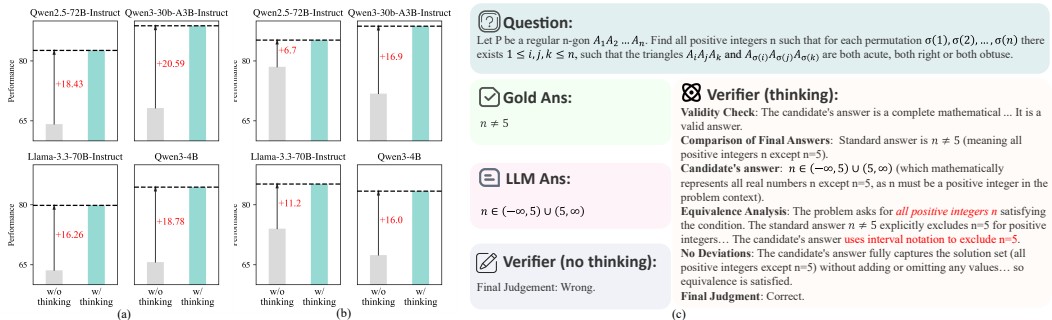

Figure 1: (a) and (b) show the performance gains with reasoning enabled on VerifierBench and VerifyBench-hard, respectively; (c) presents a case where reasoning leads to the correct judgment.

Liu et al., 2023a), but they require extensive prompt engineering, produce unstable outputs, and still struggle with complex reasoning and cross-disciplinary tasks. To overcome these limitations, we propose a solution from both the data and model perspectives, with the dual goals of establishing a systematic evaluation framework and designing a robust verifier tailored to scientific reasoning.

On the data side, we introduce SCI-VerifyBench, a cross-disciplinary and highly targeted scientific verification benchmark. We first collected over 100K responses from LLMs across mathematics, physics, biology, chemistry, and general scientific question-answering tasks, ensuring broad coverage of domains, question types, and answer formats. To increase both difficulty and specificity, we apply domain-specific equivalence transformation to some samples, enriching the test set with challenging cases. These transformations include, but are not limited to, formula rewrites, and logical equivalence substitutions, simulating error patterns that verifiers are likely to encounter in real-world scientific scenarios. Finally, by combining model-generated labels with expert human annotations, we ensured the quality and diversity of the data, making SCI-VerifyBench a benchmark that is both cross-disciplinary and rigorous for evaluating verification capabilities. In benchmark evaluations, we are surprised to find that reasoning abilities, which are often overlooked by many current specific verifiers, can significantly enhance the model's performance on scientific verification tasks.

At the model level, we propose SCI-Verifier, a reasoning-augmented verifier tailored for scientific domains. In Fig. 1, we highlight the critical role of reasoning in scientific verification. Enabling Chain-of-Thought (CoT) across models consistently boosts judgment accuracy. Motivated by this observation, SCI-Verifier adopts a two-stage post-training pipeline that combines supervised fine-tuning and reinforcement learning to incorporate logical reasoning into the verification process. This design enables it to handle complex equivalence judgments and multi-step reasoning while maintaining concise and stable outputs for deployment. It assesses equivalence from multiple perspectives while maintaining concise and stable outputs for practical deployment. Experimental results show that SCI-Verifier substantially improves accuracy on challenging and easily confusable samples compared to current verifiers and exhibits stronger cross-disciplinary generalization. Notably, the 8B version of SCI-Verifier achieves verification performance on par with the current state-of-the-art closed-source model GPT-5 (OpenAI, 2025b).

In summary, this work makes three key contributions as following:

• We propose a cross-disciplinary, high-challenge benchmark for scientific verification, SCI-VerifyBench, which covers mathematics, physics, biology, chemistry and general question-answer fields. Using real LLM responses and domain-specific equivalence transformations, it evaluates verification performance in complex scenarios and sets a unified standard for LLM assessment.

• We design a reasoning-enhanced high-performance scientific verifier, SCI-Verifier. By integrating logical reasoning via supervised post-training, SCI-Verifier gains the capability to perform complex equivalence judgments and conduct multi-step scientific reasoning, thereby significantly outperforming existing verification models across multiple domains.

• Extensive experiments show that SCI-VerifyBench and SCI-Verifier together provide a precise evaluation framework and practical guidance for improving LLM capabilities, reliability, and reasoning in scientific domain, setting a new standard for cross-disciplinary verification research.

## 2 RELATED WORKS

**Verification Benchmark.** The unstructured LLM outputs makes the verification of the answers challenging, motivating efforts to construct benchmarks for evaluating the verifiers. VAR (Chen et al., 2025a) evaluates 19 LLMs on 24 datasets to train and assess xVerify. VerifyBench (Li et al., 2025) covers general, logical, mathematical reasoning, and VerifierBench (Liu et al., 2025) has 4,000 expert-level questions across STEM domains. VerifyBench (Yan et al., 2025) aggregates model outputs with manual meta-error analysis across math, science, knowledge, and general reasoning tasks. Existing works face two main issues: (1) pointless samples, such as multiple-choice questions that require no specially designed verifiers, and (2) limited disciplinary coverage that restricts generalization assessment of scientific domain. To address these problems, we introduce SCI-VerifyBench, spanning mathematics, physics, chemistry, biology, and general QA, with filtered tasks and expert annotations to enable rigorous evaluation of scientific verification.

**Verification Models.** The verifier compensates for gaps in rule-based answer evaluation. xVerify (Chen et al., 2025a) is efficient but lacks reasoning, limiting performance; General-Verifier (Ma et al., 2025) has partial reasoning capabilities for cross-domain equivalence assessment. CompassVerifier (Liu et al., 2025) aims to provide efficient, high-performance, and robust answer verification using carefully designed error templates. Existing verifiers, constrained by limited reasoning capabilities, are inadequate for complex scientific reasoning, while using general models as verifiers requires careful prompt design with unstable outputs. Then, we propose SCI-Verifier, a reasoning-augmented scientific verifier offering strong reasoning with concise, stable outputs.

**Reward Models.** Reward models differ from verifiers in that they rank response quality, while verifiers assess correctness. Prior work primarily follows a discriminative paradigm, outputting a scalar score directly (Ouyang et al., 2022; Snell et al., 2025). More recent approaches leverage reasoning capabilities to enhance reward model performance. For example, J1 (Whitehouse et al., 2025) proposes an RL framework for training Thinking-LLM-as-a-Judge models; Think-J (Huang et al., 2025) introduces offline and online RL-based methods for judgment thinking optimization; Compass-Judger2 (Zhang et al., 2025b) uses verifiable rewards and rejection sampling to guide critical reasoning, improving robustness and generalization. Despite these advances, reward models aim to rank response quality rather than verify correctness, and this difference in objective introduces new challenges for data construction and training strategies.

## 3 SCI-VERIFYBENCH

Current research on scientific verification faces a major bottleneck in the lack of comprehensive and rigorous benchmarks, which creates blind spots in evaluating LLMs' scientific reasoning capabilities and guiding their training. To address this problem, we construct SCI-VerifyBench, a systematic cross-disciplinary benchmark covering mathematics, physics, chemistry, biology, and general scientific QA, designed to comprehensively evaluate the verification abilities of verifiers. We first present the characteristics of SCI-VerifyBench in Sec. 3.1, followed by the construction pipeline in Sec. 3.2.

Table 1: Comparison of verification benchmarks.

|  | Scale | Domains | Equivalence Transformation | Difficulty Control |
|---|---|---|---|---|
| VerifyBench | 2000 | 3 | ✗ | ✗ |
| VerifyBench-hard | 1000 | 3 | ✗ | ✓ |
| VerifierBench | 2817 | 4 | ✗ | ✗ |
| SCI-VerifyBench | 2500 | **5** | ✓ | ✓ |

### 3.1 DATA OVERVIEW

We construct SCI-VerifyBench to assess verifiers' scientific verification capabilities. In this part, we compare it with existing benchmarks and present static data analyses. Tab. 1 highlights the key differences, showing that SCI-VerifyBench spans a wider range of domains, incorporates more challenging yet commonly encountered equivalence transformations, and applies difficulty control mechanisms to better reflect realistic scientific reasoning while increasing overall task complexity. Tab. 2 presents the static analysis results, with further details provided in Appendix A.

Table 2: Benchmark Statistic.

| Statistic | Number |
|---|---|
| Total Data (each domain) | 500 |
| Real QA (each domain) | 350 |
| Synthetic QA (each domain) | 150 |
| Average Answer Tokens | 24.98 |
| Average Response Tokens | 2980.43 |

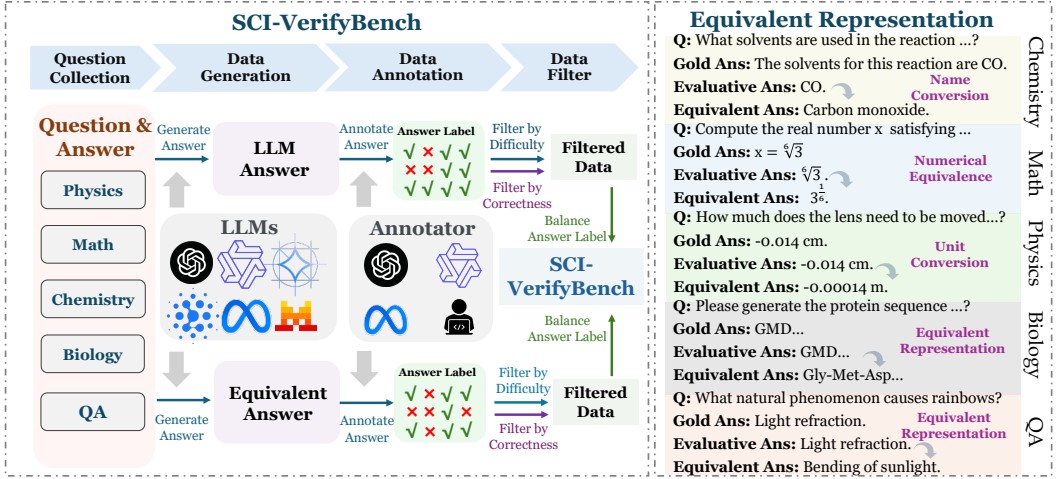

Figure 2: SCI-VerifyBench construction pipeline (left) and cases of Equivalent Representation (right).

## 3.2 DATA COLLECTION

**Question–Answer Data Generation.** Comprehensive scientific verification requires coverage of diverse question areas, answer types, and response formats. To achieve this, we collect over 15k question-answer pairs across mathematics (Gao et al., 2025), physics (Zheng et al., 2025), chemistry (Alampara et al., 2024), biology (Jiang et al., 2025), and general QA (Wang et al., 2024; Wei et al., 2024), and generate 100K+ responses using eight models of varying scales, while controlling response length. This design allows the verifier to adapt to different question and answer styles. The prompts used for data generation are provided in the Appendix A.1.

**Synthetic Data Generation.** Current verifiers perform well on responses identical to reference answers. However, even carefully trained specialized verifiers can fail when confronted with complex, domain-specific equivalence transformations, which are common in real-world scientific reasoning. Fig. 2 illustrates typical equivalence transformations across mathematics, physics, chemistry, biology, and general QA. To address this, we select 500 representative questions from each domain that allow for answer equivalence transformations and generate five equivalent answers for each using the methodology described in Appendix A.1. During this process, five LLMs assist in assessing the quality of generated equivalences. If the equivalence is clearly invalid and multiple models agree, the sample is discarded and regenerated. This approach increases the challenge for verifiers and closely simulates the diverse answer forms encountered in realistic scientific reasoning scenarios.

**Data Annotation.** The previous two pathways produced both real and synthetic question-answer data. To ensure annotation quality, we adopt a hybrid approach combining LLMs and human experts. First, five LLMs evaluate the accuracy of generated or synthetic answers against the reference answers as shown in Appendix A.2. To reduce human effort and maintain dataset difficulty, we retain only samples where the five LLMs disagree. From the data in the two methods described above, we select 2,500 samples with the highest model disagreement (500 per domain) for human annotation. Each sample is assessed by at least two experts with a bachelor's degree or higher, and answers are considered equivalent if they can be transformed into each other. In cases of disagreement, a third expert is consulted, ensuring labels reflect true equivalence while maintaining diversity.

**Data Filter.** Using the above procedure, we obtained a dataset comprising 5,000 human-annotated samples and a large collection of LLM-annotated samples. We partitioned the data into a training set and a test set, with the latter corresponding to SCI-VerifyBench. For the test set, we sampled 350 real LLM responses and 150 equivalence-based synthetic responses per domain. The selection criterion required full agreement among human experts, while samples with disagreement between human experts and LLMs are preferentially included to increase difficulty. This process results in a test set of 2,500 samples in total. The remaining non-overlapping data formed the training set, where only a portion was human-annotated and the majority relied on LLM annotations. Samples with substantial disagreement among LLMs were filtered out to ensure label reliability, yielding a training set of 14K

samples. Both the training and test sets can be expressed in the following format:

$$D = \{(q_i, a_i, r_i, l_i)\}_{i=1}^N, \tag{1}$$

where $q_i$ denotes the question, $a_i$ denotes the reference answer, $r_i$ denotes the response whose correctness needs to be evaluated, and $l_i$ denotes the label, which can be either `true` or `false`.

# 4 SCI-VERIFIER

In this section, we develop a reasoning-augmented verifier for scientific verification. First, Sec. 4.1 presents the motivation and necessity for incorporating reasoning capabilities into scientific verification. Then, Sec. 4.2 describes the approaches for integrating reasoning through supervised fine-tuning (SFT) and reinforcement learning (RL). This design emulates human step-by-step reasoning and improves verification reliability and robustness.

## 4.1 MOTIVATION

We begin by motivating the introduction of reasoning capabilities into scientific verification. While Chain-of-Thought (CoT) has been widely recognized for enhancing model performance across various domains, most existing verifier studies have overlooked this aspect. As shown in Fig. 1, we evaluate models of different scales under two conditions: outputting only the final answer versus producing intermediate reasoning before the answer. The results demonstrate that reasoning brings substantial gains in scientific verification, largely because responses of scientific questions are inherently complex and often involve multiple equivalent forms. Reasoning is thus essential for assessing equivalence from different perspectives. Building on this insight, we argue that optimizing reasoning for scientific verification can significantly boost verifier performance. Next, we introduce SCI-Verifier, a unified verifier designed to deliver concise yet powerful reasoning capabilities.

## 4.2 POST-TRAINING

In this section, we present our approach to enhancing the reasoning ability of models for scientific verification. While reasoning is essential, the nature of verification requires reasoning paths to be as concise as possible to minimize resource consumption. Accordingly, we aim for a lightweight verifier with short and stable outputs. Based on these characteristics, we utilize a two-stage post-training paradigm that combines supervised fine-tuning (SFT) with reinforcement learning (RL).

**Supervised Fine-Tuning (SFT).** In this stage, we employ large models with rejection sampling to generate a diverse set of reasoning paths in a structured format as shown in Appendix A.2. We then perform strict filtering to retain only valuable and concise traces. For reasoning models, we keep only the conclusive summary, while for non-reasoning models, we discard overly long or unstructured responses. The filtered reasoning paths are used to fine-tune a smaller model, effectively injecting the essential reasoning ability with minimal overhead. The training objective is defined as follows:

$$\mathcal{L}_{\text{SFT}}(\theta) = -\mathbb{E}_{(x,y) \sim \mathcal{D}_{\text{SFT}}}\Big[ \log \pi_\theta(y \mid x) \Big], \tag{2}$$

where $\mathcal{D}_{\text{SFT}}$ denotes the curated training dataset of high-quality reasoning traces. Unlike SFT in other domains like mathematics or physics where the focus is mostly on output formatting, verification SFT centers on transferring domain-specific verification knowledge to small models, which is essential for developing concise and useful reasoning for verification.

**Reinforcement Learning (RL).** After SFT, the model has basic verification ability and can follow the required output format, but it is prone to overfitting. we use DAPO (Yu et al., 2025), a refined GRPO (Shao et al., 2024) variant that filters both overly easy and overly hard samples and adds a length penalty to encourage concise reasoning. The training objective is to maximize:

$$\mathcal{J}_{\text{DAPO}}(\theta) = \mathbb{E}_{(q,a) \sim \mathcal{D}, \{o_i\}_{i=1}^G \sim \pi_{\theta_{\text{old}}}(\cdot \mid q)} \left[ \frac{1}{|\{o_i\}|} \sum_{i=1}^G \sum_{t=1}^{|o_i|} \right.$$

$$\left. \min\left( r_{i,t}(\theta)\, \hat{A}_{i,t},\ \text{clip}\big(r_{i,t}(\theta),\ 1 - \epsilon_{\text{low}},\ 1 + \epsilon_{\text{high}}\big)\, \hat{A}_{i,t} \right) \right] \tag{3}$$

Table 3: Performance of different verifiers on SCI-VerifyBench. Specialized verifiers use default prompts, while all other models are allowed to use reasoning.

| Models | Math | Physics | Chemistry | Biology | QA | Total | Avg. Token |
|---|---|---|---|---|---|---|---|
| Closed-source Models | | | | | | | |
| GPT-5 (OpenAI, 2025b) | 90.0 | 89.0 | 85.4 | 84.8 | **95.4** | 88.92 | 384.59 |
| Gemini-2.5-Flash (Comanici et al., 2025) | 90.2 | 81.8 | 86.0 | 88.2 | 86.6 | 87.56 | 478.48 |
| o4-mini (OpenAI, 2025a) | 88.8 | 84.0 | 84.6 | 86.6 | 90.8 | 86.96 | 437.27 |
| Open-source Instruct models | | | | | | | |
| Qwen2.5-72B-Instruct (Qwen et al., 2025) | 90.2 | 80.6 | 77.8 | 80.6 | 82.0 | 82.24 | 400.40 |
| Qwen3-30B-A3B-Instruct-2507 (Qwen et al., 2025) | 88.4 | 80.0 | 79.6 | 88.4 | 85.6 | 84.40 | 684.58 |
| LLaMa-3.3-70B-Instruct (Grattafiori et al., 2024) | 76.6 | 67.4 | 78.8 | 84.6 | 85.8 | 78.64 | 364.36 |
| Open-source Reasoning Models | | | | | | | |
| Qwen3-4B (Yang et al., 2025) | 88.2 | 77.2 | 81.4 | 83.8 | 83.4 | 82.80 | 1466.92 |
| Qwen3-8b (Yang et al., 2025) | 90.0 | 76.2 | 81.8 | 83.0 | 81.4 | 82.48 | 1033.07 |
| GPT-oss-20b (OpenAI et al., 2025) | 85.0 | 70.0 | 81.0 | 84.6 | 72.4 | 78.60 | 522.55 |
| Qwen3-30B-A3B-Thinking-2507 (Yang et al., 2025) | 82.2 | 70.4 | 86.0 | 88.4 | 84.8 | 82.36 | 1714.66 |
| GPT-oss-120B (OpenAI et al., 2025) | 79.8 | 69.8 | 86.2 | 87.0 | 90.0 | 82.66 | 110.21 |
| Qwen3-235B-A22B (Yang et al., 2025) | 82.4 | 67.0 | **87.8** | 91.2 | 79.8 | 81.64 | 4601.16 |
| Specific Verifiers | | | | | | | |
| xVerify-8B (Chen et al., 2025a) | 77.8 | 60.6 | 85.8 | 88.6 | 88.0 | 80.16 | 1.00 |
| CompassVerifier-3B (Liu et al., 2025) | 86.2 | 79.4 | 80.2 | 86.6 | 87.0 | 83.88 | 192.00 |
| CompassVerifier-7B (Liu et al., 2025) | 87.4 | 82.0 | 84.0 | 84.6 | 87.2 | 85.04 | 162.34 |
| CompassVerifier-32B (Liu et al., 2025) | 90.0 | 82.0 | 84.0 | 85.4 | 89.8 | 86.24 | 212.04 |
| Ours | | | | | | | |
| SCI-Verifier-4B | 92.4 | 84.6 | 86.4 | 94.2 | 93.4 | 90.20 | 485.13 |
| SCI-Verifier-8B | **93.8** | **90.4** | 87.8 | **96.4** | 95.2 | 92.72 | 490.66 |

where $r_{i,t}(\theta) = \frac{\pi_\theta\left(o_{i,t}|q,o_{i,<t}\right)}{\pi_{\theta_{\text{old}}}\left(o_{i,t}|q,o_{i,<t}\right)}$. Here, $\pi_\theta$ and $\pi_{\theta_{\text{old}}}$ denote the updated and previous policies.

The advantage function $\hat{A}_{i,t}$ is calculated from the final reward $R_i$:

$$\hat{A}_{i,t} = \frac{R_i - \text{mean}(\{R_j\}_{j=1}^G)}{\text{std}(\{R_j\}_{j=1}^G)} \tag{4}$$

The final reward $R_i$ is a sum of the alignment reward $R_{\text{align},i}$ and a overlong penalty $P_{\text{overlong},i}$:

$$R_i = R_{\text{align},i} + P_{\text{overlong},i} \tag{5}$$

where $R_{\text{align},i}$ is 1 if the final prediction for example $i$ matches the ground-truth answer, and 0 otherwise, the overlong penalty is defined as:

$$P_{\text{overlong},i} = \begin{cases} 0 & \text{if } |o_i| \leq L_{\text{max}} \\ -\frac{|o_i|-L_{\text{max}}}{L_{\text{buffer}}} \cdot \lambda_{\text{penalty}} & \text{if } L_{\text{max}} < |o_i| \leq L_{\text{max}} + L_{\text{buffer}} \\ -\lambda_{\text{penalty}} & \text{if } |o_i| > L_{\text{max}} + L_{\text{buffer}} \end{cases} \tag{6}$$

Here, $|o_i|$ is the length of the response, $L_{\text{max}}$ is the maximum allowed length, $L_{\text{buffer}}$ is the overlong buffer length, and $\lambda_{\text{penalty}}$ is the penalty weight. Since verification is a binary classification task, imbalanced data may lead the model to rely on label priors instead of reasoning. To address this, we rebalance the dataset during RL training to ensure equal positive and negative examples.

Through this two-stage post-training paradigm, we obtain a verification model with concise reasoning ability, applicable across domains for scientific answer validation. This approach not only improves the reliability of model capability evaluation, but also provides a reward function with clear semantics, thereby facilitating the training of stronger reasoning-oriented language models.

# 5 EXPERIMENTS

## 5.1 BASELINES AND SETUP

We conduct a systematic evaluation of SCI-VerifyBench on SCI-Verifier-4B and 8B, which are trained from Qwen3-4B-Base (Yang et al., 2025) and Qwen3-8B-Base (Yang et al., 2025), respectively. In addition, we benchmark on two established datasets: VerifierBench (Liu et al., 2025) and VerifyBench-hard (Yan et al., 2025). The baselines cover four categories: (1) closed-source models, (2) open-source instruct models, (3) open-source reasoning models, and (4) specialized verifiers. Details are provided in Appendix A.2. For evaluation, we report Accuracy on SCI-VerifyBench, since positive and negative samples are balanced by construction. On VerifierBench and VerifyBench-hard, we additionally report F1 score alongside Accuracy. In all cases, higher values indicate stronger verification performance.

Table 4: Performance of different verifiers on VerifierBench and VerifyBench-Hard. Specialized verifiers use default prompts, while all other models are allowed to use reasoning.

| Models | VerifierBench | | | VerifyBench-Hard | | |
|---|---|---|---|---|---|---|
| | Acc. | F1 | Avg. Token | Acc. | F1 | Avg. Token |
| Closed-source Models | | | | | | |
| GPT-5 (OpenAI, 2025b) | 91.80 | 90.48 | 203.45 | 90.40 | 85.34 | 245.64 |
| Gemini-2.5-Flash (Comanici et al., 2025) | 87.63 | 87.56 | 265.49 | 87.70 | 83.65 | 302.65 |
| Open-source Instruct models | | | | | | |
| Qwen2.5-72B-Instruct (Qwen et al., 2025) | 82.61 | 81.67 | 550.73 | 85.20 | 81.31 | 381.27 |
| Qwen3-30B-A3B-Instruct-2507 (Qwen et al., 2025) | 88.78 | 88.88 | 972.30 | 88.70 | 85.03 | 810.24 |
| LLaMa-3.3-70B-Instruct (Grattafiori et al., 2024) | 79.84 | 79.00 | 398.99 | 85.20 | 81.10 | 382.04 |
| Open-source Reasoning Models | | | | | | |
| Qwen3-4B (Yang et al., 2025) | 84.42 | 84.54 | 2119.87 | 83.40 | 78.80 | 1755.61 |
| Qwen3-8b (Yang et al., 2025) | 85.55 | 85.56 | 1857.95 | 84.40 | 79.58 | 1588.45 |
| GPT-oss-20B (OpenAI et al., 2025) | 83.36 | 83.73 | 523.10 | 85.90 | 80.36 | 328.50 |
| Qwen3-30B-A3B-Thinking-2507 (Yang et al., 2025) | 90.42 | 90.05 | 2438.52 | 88.60 | 84.92 | 2226.46 |
| Qwen3-235B-A22B (Yang et al., 2025) | 88.36 | 88.01 | 5044.43 | 86.80 | 82.26 | 4690.13 |
| Specific Verifiers | | | | | | |
| xVerify-8B (Chen et al., 2025a) | 78.03 | 75.53 | 1.00 | 83.20 | 79.60 | 1.00 |
| CompassVerifier-3B (Liu et al., 2025) | 82.39 | 83.37 | 1.00 | 86.60 | 84.16 | 1.00 |
| CompassVerifier-7B (Liu et al., 2025) | 85.56 | 84.83 | 1.00 | 87.50 | 84.13 | 1.00 |
| CompassVerifier-32B (Liu et al., 2025) | 89.88 | 88.91 | 1.00 | 88.30 | 85.86 | 1.00 |
| Ours | | | | | | |
| SCI-Verifier-4B | 92.37 | 92.01 | 703.47 | 88.90 | 85.98 | 470.26 |
| SCI-Verifier-8B | **93.01** | **93.06** | 636.53 | **90.30** | **87.45** | 393.61 |

## 5.2 EVALUATION AND ANALYSIS OF SCI-VERIFYBENCH

In this part, we present and analyze the evaluation results on SCI-VerifyBench. Tab. 3 reports the performance of both closed-source and open-source models on SCI-VerifyBench. This comprehensive evaluation enables us to compare the verification capabilities of LLMs of different types and scales under the same settings. We then provide a detailed analysis of the experimental results.

**Open-source models are gradually closing the gap with proprietary models, yet a noticeable performance gap remains.** On the verification task, many open-source models have approached the performance of closed-source models, including specialized verifiers, but proprietary models still maintain an edge. For instance, GPT-5 outperforms current open-source models by more than 5%. Notably, our proposed SCI-Verifier achieves performance comparable to GPT-5 on the scientific verification task, which confirms the effectiveness of the proposed verifier.

**Reasoning models and chat models do not exhibit significant differences on this task.** On this task, reasoning models show no clear advantage over chat models. We attribute this to the fact that, unlike challenging problems such as IMO-level mathematics, scientific verification tasks are straightforward, requiring domain-specific knowledge and only brief reasoning. Since both model types share similar priors and lack reasoning-specific optimization, performance gains are limited. This observation underscores the need for reasoning tailored to the unique characteristics of verification tasks.

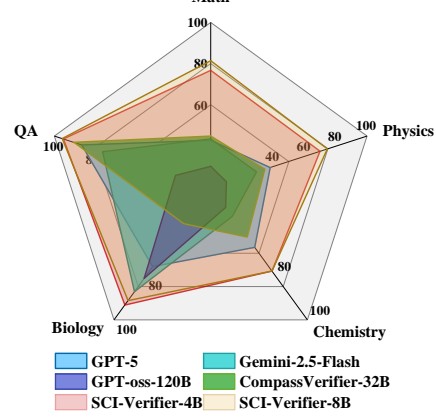

Figure 3: Evaluation on Equivalent Answer.

**Equivalence-based answers poses significant challenges for current LLMs.** As shown in Fig. 3, on our equivalence-augmented test set derived from SCI-VerifyBench, even state-of-the-art GPT-5 models perform poorly, with scores dropping below 60% in mathematics and physics. This highlights a clear deficiency in handling complex equivalence transformations. Remarkably, our SCI-Verifier, in both its 4B and 8B configurations, achieves substantially higher performance on the same tests, owing to targeted optimization for this challenge. These results provide strong evidence for the effectiveness of integrating reasoning capabilities specifically tailored for equivalence verification.

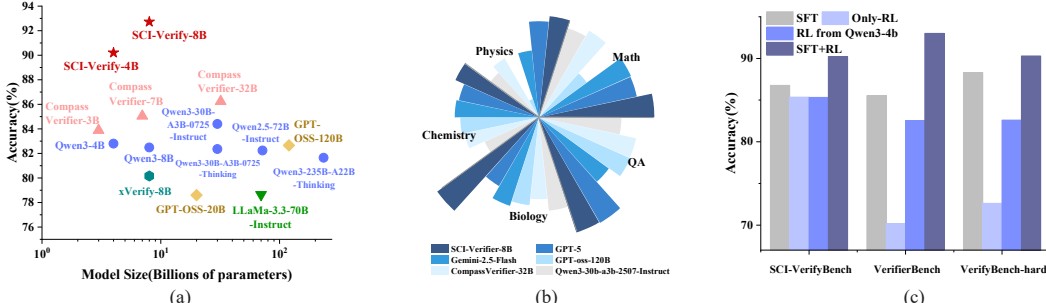

Figure 4: (a) Performance on SCI-VerifyBench versus model size. (b) Difficulty comparison across domains in SCI-VerifyBench. (c) Ablation study of training methods.

**Model scale does not have a decisive impact on results.** Experiments across model scales show that increasing model size does not consistently improve performance (Fig. 4(a)). We attribute this to the fact that verification mainly relies on prior knowledge for answer equivalence, and current models are not specifically optimized for this objective, so larger capacity does not yield better performance.

**Task characteristics across domains lead to domain-dependent performance differences.** As shown in Tab. 3 and Fig. 4(b), performance varies across disciplines but follows consistent trends across models. Mathematics and physics achieve lower scores due to complex transformations such as Taylor expansion, whereas other subjects involve more straightforward judgments once prerequisite knowledge is available. These findings suggest the need for discipline-specific verifiers.

## 5.3 EVALUATION AND ANALYSIS OF SCI-VERIFIER

**Generalization of SCI-Verifier.** We conduct experiments on our SCI-VerifyBench and two existing verification benchmarks, VerifierBench and VerifyBench-Hard as shown in Tab. 4. The results demonstrate that, both sizes of SCI-Verifier achieve strong performance even at small sizes, reaching levels comparable to the state-of-the-art closed-source model GPT-5. Meanwhile, Fig. 3 demonstrates the strong capability of SCI-Verifier in judging equivalence transformations. The consistent advantage of SCI-Verifier across all three benchmarks indicates its strong verification ability and generalization capability across tasks. Notably, on SCI-VerifyBench, SCI-Verifier outperforms current open-source models in all disciplines, further validating its cross-disciplinary generalization in verification.

**Prompt Robustness of SCI-Verifier.** We investigate the robustness of SCI-Verifier to prompt variations, a property that is crucial for real-world applications where prompts must often be adapted to user requirements (Liu et al., 2023b). We evaluate multiple models on three benchmarks using both our proposed CoT prompt and the xVerify prompt (modified to allow reasoning for alignment pur-

Table 5: Comparison of model robustness across different prompts. *our*: default prompt; *other*: modified prompt.

| Models | SCI-VerifyBench | | VerifyBench-Hard | |
|---|---|---|---|---|
| | our | other | our | other |
| Qwen3-30B-A3B-Instruct-2507 | 84.40 | 82.92 | 88.70 | 75.40 |
| GPT-oss-20b | 78.60 | 79.08 | 85.90 | 79.50 |
| Qwen3-235B-A22B | 81.64 | 79.40 | 86.80 | 81.00 |
| CompassVerifier-3B | 83.88 | 81.72 | 86.60 | 79.30 |
| CompassVerifier-7B | 85.04 | 84.84 | 87.50 | 80.30 |
| CompassVerifier-32B | 85.04 | 86.00 | 88.30 | 84.70 |
| SCI-Verifier-4B | 90.20 | 89.80 | 88.90 | 88.30 |
| SCI-Verifier-8B | 92.72 | 91.90 | 90.30 | 89.70 |

poses). Details are provided in Appendix A.2, and the results are summarized in Tab. 5. From these results, we draw two key conclusions: (1) SCI-Verifier exhibits strong robustness to prompt modifications, maintaining competitive performance even when the prompt differs from those seen during training; and (2) general models are considerably more sensitive to prompt variations in verification tasks, largely because they lack an intrinsic notion of answer equivalence and must instead rely on contextual cues. Notably, this sensitivity tends to diminish as model size increases.

## 5.4 ABLATION STUDY

**Training Methods.** In this section, we analyze the contribution of each component in our two-stage training framework. Experiments on both 4B and 8B models (Fig. 5(a)) show that SFT alone

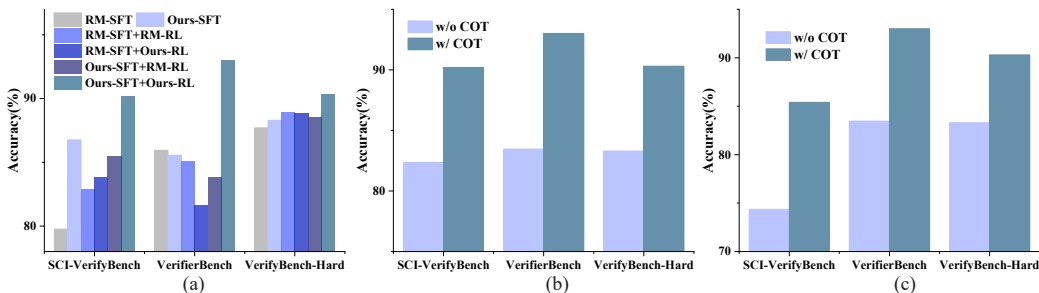

Figure 5: (a) Ablation study of training data impact. (b) Ablation study of SFT distillation methods. (c) Ablation study of training with CoT in scientific verification.

already yields strong verification performance, indicating that supervised adaptation effectively equips the model with basic task-specific reasoning ability. Starting RL from a reasoning model also achieves competitive results, whereas directly applying RL to the Base model performs poorly, likely because the absence of SFT warm-up limits the acquisition of targeted reasoning skills. In contrast, combining SFT and RL consistently delivers the best overall performance, particularly in cross-dataset generalization, demonstrating that the two stages play complementary roles and are both crucial to the final effectiveness.

**Training Data.** In this part, we evaluate the quality of our constructed training dataset by comparing it with a commonly used dataset (RM) (Zhao et al., 2025) in the Reward Model domain. Using Qwen3-4B-Base as the initial model, we conduct experiments with both datasets under SFT and SFT+RL settings, and the detailed results are presented in Fig. 5(b). The results show that our dataset consistently enables the model to achieve strong performance across three benchmarks, whether used for SFT or RL. This demonstrates the high quality of our data, from which the model can learn richer distributional information about the verification task. The RM dataset also yields reasonable performance under SFT, mainly because of its large scale with more than 180K samples. However, its effectiveness under RL is limited since the heterogeneous quality within such a large dataset slows down model improvement, which makes data filtering necessary in practice. These findings confirm that our constructed training dataset, like our test data, is of high quality and reliability.

**Distillation Data.** We investigate the effectiveness of our proposed short CoT distillation. Specifically, we compare the outcomes of distilling complete CoT versus short CoT, with results presented in Fig. 5(b). The findings reveal that distilling complete CoT not only fails to improve performance but also substantially increases output length, rendering it impractical. We attribute this to the nature of the verification task, which is relatively simple and does not require long reasoning chains. Instead, concise reasoning from fixed perspectives is sufficient to achieve strong performance. Therefore, distilling short reasoning traces during the SFT stage is both a reasonable and efficient choice.

**Inference Mode.** In this part, we investigate the impact of incorporating reasoning capabilities on model performance. We compare models trained with and without reasoning modes using the same training data, with results shown in Fig. 5(c). We find that omitting chain-of-thought leads to more efficient inference but results in a substantial performance drop. This clearly demonstrates the importance of incorporating reasoning abilities for verification tasks in scientific domains.

## 6 CONCLUSION

We highlight verification as a critical step toward advancing the scientific reasoning capabilities of LLMs. To this end, we introduce SCI-VerifyBench, a high-quality and diverse benchmark spanning mathematics, physics, chemistry, biology, and commonsense scientific QA tasks, designed to rigorously and systematically assess models' cross-disciplinary scientific verification capabilities. Our study further demonstrates that chain-of-thought reasoning is essential for scientific verification, particularly when answers are complex or admit multiple equivalent forms. Building on this insight, we develop SCI-Verifier, a verifier endowed with concise reasoning abilities specifically tailored for verification tasks. Together, SCI-VerifyBench and SCI-Verifier provide both a comprehensive evaluation framework and a practical solution for scientific verification, offering strong potential to guide the continued advancement and reliability of LLMs in scientific reasoning.

## ACKNOWLEDGMENTS

This work was supported by the Shanghai Artificial Intelligence Laboratory and a locally commissioned task from the Shanghai Municipal Government.

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

# Appendix for SCI-Verifier

## A  DETAILS OF SCI-VERIFYBENCH

In this section, we introduce details of the process of constructing SCI-VerifyBench, including the prompts and models used for data generation described in Sec. A.1, the model annotation process and the prompts and parameters designed for practical use described in Sec. A.2, as well as the data details and sample cases in SCI-VerifyBench described in Sec. A.3.

### A.1  DATA GENERATION

The data generation involves two parts. The first part uses LLMs to generate answers for existing questions, and correctness is determined by comparing the generated answers with the reference answers. The second part generates equivalent answers based on the characteristics of different subjects, testing whether the model can correctly recognize these equivalent forms. For both parts, multiple LLMs are used to generate candidate answers, including Qwen3-32B, Qwen3-30B-A3B-Thinking-2507, Qwen3-30B-A3B-Instruct-2507, LLaMa3.3-70B-Instruct, GPT-oss-20B, Qwen2.5-32B-Instruct, Gemma-3-27b-it, and Qwen3-8B. The prompt used in the first part is shown in Box. A.1. For the second part, different prompts are used for each subject according to the corresponding task. The Math prompts refer to Box. A.2 to Box. A.4. The Physics prompts refer to Box. A.5 to Box. A.7. The Chemistry prompts refer to Box. A.8 to Box. A.16. The Biology prompts refer to Box. A.17 to Box. A.21. The QA prompts refer to Box. A.22.

---

**Box A.1: Prompt for generating LLM response**

Please answer the problem adhering to the following rules: 1. Please use LaTeX format to represent the variables and formulas used in the solution process and results. 2. Please put the final answer(s) in boxed{}, note that the unit of the answer should not be included in boxed{}. 3. If the problem requires multiple answers, list them in order, each in a separate boxed{}. Problem:{answer}

---

**Box A.2: Prompt for generating equivalent answers to mathematical interval problems**

You are given a mathematical interval answer. Generate 10 different equivalent forms of this interval, using transformations such as:
- Rewriting as inequalities: $[0, 1] \rightarrow 0 \leq x \leq 1$
- Rewriting as set operations: $[0, 2] \cup [1, 3] = [0, 3]$
- Open/closed interval limit definitions: $(a, b) = \lim_{\epsilon \to 0^+} [a + \epsilon, b - \epsilon]$
- Converting to numeric sets: $[0, 2] \rightarrow \{0, 1, 2\}$ (for integer endpoints)

**Format**: Output exactly 10 forms. Each form must be wrapped in LaTeX \boxed{...}. Separate each answer with a newline (\n).
**Example**:
Input: $[0, 1]$
Output:

$$\boxed{0 \leq x \leq 1}$$

$$\boxed{[0, 1) \cup 1}$$

$$\boxed{(0 - 0.0001, 1 + 0.0001)}$$

$$\boxed{x \in \mathbb{R} \mid 0 \leq x \leq 1}$$

$$\boxed{0, 0.25, 0.5, 0.75, 1}$$

Only output the converted results, do not output the conversion process!
Now, process the following answer: {answer},

---

---

**Box A.3: Prompt for generating equivalent answers to mathematical expression problems**

You are given a mathematical expression or a numeric answer. Generate 10 different equivalent forms of this expression, using transformations such as:

- Factoring or expansion: $x^2 + 2x + 1 \rightarrow (x+1)^2$
- Fraction simplification: $(x^2 - 1)/(x+1) \rightarrow x - 1$
- Leaving fraction unsimplified: $(x^2 - 1)/(x+1)$ (unchanged)
- Partial fraction decomposition: $1/(x(x+1)) \rightarrow 1/x - 1/(x+1)$
- Fraction to decimal conversion: $1/2 \rightarrow 0.5$
- Trigonometric identities: $\sin^2 x + \cos^2 x = 1$
- Trigonometric transformations: $\sin 2x = 2 \sin x \cos x$
- Taylor expansion: $\sin x \approx x - x^3/3!$
- Exponential/logarithm rules: $ln(ab) = lna + lnb$
- Substitution: let y=x+1, then $x^2 + 2x + 1 = y^2$
- Approximating special constants: $\pi \approx 3.14159$, $e \approx 2.718$
- Angle-radian conversion: $\pi/6 = 30°$

**Format**: Output exactly 10 forms. Each form must be wrapped in LaTeX \boxed{...}. Separate each answer with a newline (\n).

**Example**:

  Input: $(x+1)^2$

  Output:

  $\boxed{(x+1)(x+1)}$

  $\boxed{x^2 + 2x + 1}$

  $\boxed{\dfrac{(x+1)^3}{x+1}}$

  $\boxed{\text{Let } y = x+1, ; y^2}$

  $\boxed{(x+1)^2 \approx 1 + 2x; \text{ for small } x}$

Only output the converted results, do not output the conversion process!

Now, process the following answer: {answer},

---

---

**Box A.4: Prompt for generating equivalent answers to mathematical equation problems**

You are given a mathematical equation. Generate 10 different equivalent forms of this equation, using transformations such as:
- Moving terms: $x + 3 = 5 \rightarrow x = 2$
- Multiplying/dividing both sides: $2x = 4 \rightarrow x = 2$
- Factoring: $x^2 - 1 = 0 \rightarrow (x - 1)(x + 1) = 0$
- Completing the square: $x^2 + 6x + 5 = 0 \rightarrow (x + 3)^2 - 4 = 0$
- Root extraction: $x^2 = 4 \rightarrow x = \pm 2$
- Substitution in equations: $x^2 + 1 = 0 \rightarrow let\, x = i, then\, i^2 + 1 = 0$
- Trigonometric transformations: $\sin^2 x = 1 - \cos^2 x$
- Taylor expansion for approximate solution: $\sin x \approx x \rightarrow x \approx 0$
- Domain restrictions: $\sqrt{x - 1} = x - 3$ requires $x \geq 1$

**Format**: Output exactly 10 forms. Each form must be wrapped in LaTeX \boxed{...}.
Separate each answer with a newline (\n).
**Example**:
  Input: $x^2 - 1 = 0$
  Output:

$\boxed{x^2 = 1}$

$\boxed{(x - 1)(x + 1) = 0}$

$\boxed{x = \pm 1}$

$\boxed{(x - 0)^2 - 1 = 0}$

$\boxed{\cos^2 \theta - \sin^2 \theta = 0 \quad (\text{substitution } x = \cos \theta)}$

Only output the converted results, do not output the conversion process!
Now, process the following answer: {answer},

---

**Box A.5: Prompt for generating equivalent answers to physics numerical problems**

Given the following question and an answer in expression form, generate 10 different equivalent forms of this interval, using transformations such as:
- Arithmetic operations or evaluation (e.g., $5 + 3 \rightarrow 8$)
- Substitution of variable values (e.g., $2x + 3, x = 2 \rightarrow 7$)
- Scientific notation, fractions, or decimals (e.g., $0.00012 \rightarrow 1.2 \times 10^{-4}$)
- Unit conversion or dimensional adjustment (e.g., $1km \rightarrow 1000m$)
- Dimensional conversion (e.g., $1N \rightarrow 1kg \cdot m/s^2$)

**Format**: Output exactly 10 forms. Each form must be wrapped in LaTeX \boxed{...}.
Separate each answer with a newline (\n).
**Example**:
  Input: [0,1]
  Output:

$\boxed{0 \leq x \leq 1}$

$\boxed{[0, 1) \cup 1}$

$\boxed{(0 - 0.0001, 1 + 0.0001)}$

$\boxed{x \in \mathbb{R} \mid 0 \leq x \leq 1}$

$\boxed{0, 0.25, 0.5, 0.75, 1}$

Only output the converted results, do not output the conversion process!
Now, process the following answer: Question: {question}. Answer (Numeric): {answer},

---

**Box A.6: Prompt for generating equivalent answers to physics expression problems**

Given the following question and an answer in expression form, generate 10 different equivalent forms of this interval, using transformations such as:

- Factoring or expansion: $x^2 + 2x + 1 \rightarrow (x+1)^2$
- Fraction simplification: $(x^2 - 1)/(x + 1) \rightarrow x - 1$
- Leaving fraction unsimplified: $(x^2 - 1)/(x + 1)$ (unchanged)
- Partial fraction decomposition: $1/(x(x + 1)) \rightarrow 1/x - 1/(x + 1)$
- Fraction to decimal conversion: $1/2 \rightarrow 0.5$
- Trigonometric identities: $\sin^2 x + \cos^2 x = 1$
- Trigonometric transformations: $\sin 2x = 2 \sin x \cos x$
- Taylor expansion: $sinx \approx x - x^3/3!$
- Exponential/logarithm rules: $ln(ab) = lna + lnb$
- Substitution: let $y = x + 1$, then $x^2 + 2x + 1 = y^2$
- Approximating special constants: $\pi \approx 3.14159, e \approx 2.718$
- Angle-radian conversion: $\pi/6 = 30$
- Dimensional conversion (e.g., $1N \rightarrow 1kg \cdot m/s^2$)

**Format**: Output exactly 10 forms. Each form must be wrapped in LaTeX \boxed{...}. Separate each answer with a newline (\n).

**Example**:

Input: $(x + 1)^2$

Output:

$\boxed{(x + 1)(x + 1)}$

$\boxed{x^2 + 2x + 1}$

$\boxed{\dfrac{(x+1)^3}{x+1}}$

$\boxed{\text{Let } y = x + 1; y^2}$

$\boxed{(x + 1)^2 \approx 1 + 2x; \text{for small } x}$

Only output the converted results, do not output the conversion process!

Now, process the following answer: Question: {question}. Answer (expression): {answer},

---

---

**Box A.7: Prompt for generating equivalent answers to physics equation problems**

Given the following question and an answer in expression form, generate 10 different equivalent forms of this interval, using transformations such as:

- Moving terms: $x + 3 = 5 \rightarrow x = 2$
- Multiplying/dividing both sides: $2x = 4 \rightarrow x = 2$
- Factoring: $x^2 - 1 = 0 \rightarrow (x-1)(x+1) = 0$
- Completing the square: $x^2 + 6x + 5 = 0 \rightarrow (x+3)^2 - 4 = 0$
- Root extraction: $x^2 = 4 \rightarrow x = \pm 2$
- Substitution in equations: $x^2 + 1 = 0 \rightarrow let\, x = i, then\, i^2 + 1 = 0$
- Trigonometric transformations: $\sin^2 x = 1 - \cos^2 x$
- Taylor expansion for approximate solution: $\sin x \approx x \rightarrow x \approx 0$
- Domain restrictions: $\sqrt{(x-1)} = x - 3$ requires $x \geq 1$
- Dimensional conversion (e.g., $F = ma, m = 1000g, a = 2m/s^2 \rightarrow F = 2N$)

**Format**: Output exactly 10 forms. Each form must be wrapped in LaTeX \boxed{...}. Separate each answer with a newline (\n).

**Example**:

Input: $x^2 - 1 = 0$

Output:

$\boxed{x^2 = 1}$

$\boxed{(x-1)(x+1) = 0}$

$\boxed{x = \pm 1}$

$\boxed{(x-0)^2 - 1 = 0}$

$\boxed{\cos^2 \theta - \sin^2 \theta = 0 \quad (\text{substitution } x = \cos \theta)}$

Only output the converted results, do not output the conversion process!

Now, process the following answer: Question: {question}. Answer (equation): {answer}

---

---

**Box A.8: Prompt for generating equivalent answers to chemical solvent prediction problems**

Given a chemical reaction and a predicted solvent, generate 10 equivalent forms of the solvent answer, considering:

- Different chemical names (IUPAC, common, or trivial)
- Abbreviations (e.g., EtOH)
- Molecular formulas (e.g., C2H5OH)
- SMILES representations
- Solvent class equivalence (e.g., polar protic, polar aprotic)
- Mixture equivalences (e.g., EtOH:H2O 1:1 ≡ H2O:EtOH 1:1)

**Format**: Output exactly 10 forms, each wrapped in LaTeX \boxed{...} and separated by newline.

**Example**:

  Input: Ethanol
  Output:

$\boxed{\text{Ethanol}}$

$\boxed{\text{EtOH}}$

$\boxed{\text{C2H5OH}}$

$\boxed{\text{CCO}}$

$\boxed{\text{alcohol}}$

$\boxed{\text{polar protic solvent}}$

$\boxed{\text{EtOH:H2O 1:1}}$

$\boxed{\text{H2O:EtOH 1:1}}$

$\boxed{\text{ethyl alcohol}}$

$\boxed{\text{C-C-O}}$

Only output the converted results, do not output the conversion process!

Now process: Reaction: {question} Predicted solvent: {answer},

---

---

**Box A.9: Prompt for generating equivalent answers to chemical property prediction problems**

Given a molecule and a predicted property value, generate 10 equivalent forms, using:
- Unit conversions (e.g., Celsius ↔ Kelvin ↔ Fahrenheit)
- Ranges vs single values
- Approximate vs exact
- Different notations (e.g., logP, Kow)
- Descriptive labels (e.g., high solubility)

**Format**: 10 forms wrapped in LaTeX \boxed{...}.

**Example**:

Input: Water boiling point = 100 °C

Output:

$\boxed{100\ °\text{C}}$

$\boxed{373\ \text{K}}$

$\boxed{212°\text{F}}$

$\boxed{100 - 101\ °\text{C}}$

$\boxed{\text{high boiling point}}$

$\boxed{\text{approx. } 100°\text{C}}$

$\boxed{373.15\text{K}}$

$\boxed{0.1 \times 10^3\ °\text{C}}$

$\boxed{100\ \text{Celsius}}$

$\boxed{\text{water boils at } 100°\text{C}}$

Only output the converted results, do not output the conversion process! Now process: Molecule: {question} Property: {answer},

---

---

**Box A.10: Prompt for generating equivalent answers to chemical yield prediction problems**

---

Given a chemical reaction and a predicted yield, generate 10 equivalent forms:
  - Percentage ↔ decimal
  - Ranges ↔ single value
  - Descriptive labels (e.g., high yield, trace)
  - Approximate expressions

**Format**: 10 forms, LaTeX \boxed{...}, newline separated.

**Example**:
  Input: 85%
  Output:

$\boxed{85\%}$

$\boxed{0.85}$

$\boxed{\text{high yield}}$

$\boxed{80 - 90\%}$

$\boxed{\text{approx. } 85\%}$

$\boxed{y = 0.85}$

$\boxed{\text{yield} > 80\%}$

$\boxed{\text{quantitative yield}}$

$\boxed{\text{major product}}$

$\boxed{\text{trace product}}$

Only output the converted results, do not output the conversion process!

Now process: Reaction: {question} Predicted yield: {answer},

---

---

**Box A.11: Prompt for generating equivalent answers to chemical retrosynthesis problems**

Given a target molecule, generate 10 equivalent retrosynthesis answers:
- Different valid disconnections
- Different reagents or reducing / oxidizing agents
- Alternative synthetic routes (single-step or multi-step)
- Protecting group alternatives
- Functional group or stereochemical equivalence

**Format**: 10 forms wrapped in LaTeX \boxed{...}.

**Example**:

Input: Target = Benzyl alcohol

Output:

$\boxed{\text{Benzaldehyde + NaBH4} \rightarrow \text{Benzyl alcohol}}$

$\boxed{\text{Benzyl chloride + NaOH} \rightarrow \text{Benzyl alcohol}}$

$\boxed{\text{Benzaldehyde reduced by LiAlH4}}$

$\boxed{\text{Benzyl bromide + KOH} \rightarrow \text{Benzyl alcohol}}$

$\boxed{\text{Benzaldehyde} \rightarrow \text{H2 + Pd/C} \rightarrow \text{Benzyl alcohol}}$

$\boxed{\text{C6H5CH2Cl + NaOH} \rightarrow \text{C6H5CH2OH}}$

$\boxed{\text{Alternative protecting group: Boc route}}$

$\boxed{\text{Multi-step oxidation-reduction route}}$

$\boxed{\text{Reductive amination route to same alcohol}}$

$\boxed{\text{Electrochemical reduction route}}$

Only output the converted results, do not output the conversion process!

Now process: Target molecule: {question} Retrosynthesis answer: {answer},

---

**Box A.12:  Prompt for generating equivalent answers to chemical temperature prediction problems**

---

Given a reaction and predicted temperature, generate 10 equivalent forms:
- Different units (°C, K, °F)
- Ranges vs single value
- Descriptive labels (low, room temp, high)
- Common lab descriptions (ice bath, reflux)

**Format**: LaTeX \boxed{...}, 10 forms.

**Example**:

  Input: 80°C

  Output:

$$\boxed{80\ {}^\circ\text{C}}$$

$$\boxed{353\ \text{K}}$$

$$\boxed{176\ {}^\circ\text{F}}$$

$$\boxed{70 - 90\ {}^\circ\text{C}}$$

$$\boxed{\text{approx. } 80{}^\circ\text{C}}$$

$$\boxed{\text{reflux}}$$

$$\boxed{\text{room temperature}}$$

$$\boxed{\text{ice bath}}$$

$$\boxed{\text{moderate heating}}$$

$$\boxed{\text{high temperature}}$$

Only output the converted results, do not output the conversion process!

Now process: Reaction: {question} Predicted temperature: {answer},

**Box A.13: Prompt for generating equivalent answers to chemical product prediction problems**

Given reactants and reaction conditions, generate 10 equivalent forms of the predicted product:

- Different structure representations: SMILES, InChI, molecular formula
- IUPAC name, common name, trivial name
- Stereoisomers (R/S)
- Tautomers, salts, hydrates
- Alternative valid representations (chair/boat conformers)

**Format**: 10 LaTeX \boxed{...} outputs.

**Example**:

Input: Nitrobenzene

Output:

$\boxed{\text{C6H5NO2}}$

$\boxed{\text{c1ccc(cc1)[N+](=O)[O-]}}$

$\boxed{\text{Nitrobenzene}}$

$\boxed{\text{Benzene nitro compound}}$

$\boxed{\text{C6H5-NO2}}$

$\boxed{\text{Nitrobenzol}}$

$\boxed{\text{C6H5NO2 · H2O}}$

$\boxed{\text{aromatic nitro compound}}$

$\boxed{\text{benzene derivative}}$

$\boxed{\text{Racemic mixture if applicable}}$

Only output the converted results, do not output the conversion process!

Now process: Reactants: {question} Predicted product: {answer},

**Box A.14: Prompt for generating equivalent answers to chemical mol2caption problems**

Given a molecular structure, generate 10 equivalent textual descriptions:
- IUPAC name
- Common/trivial names
- Chinese name
- Functional description (solvent, reagent, drug)
- Usage or property description

**Format**: LaTeX \boxed{...}, 10 forms.
**Example**:
  Input: C2H5OH
  Output:

  | Ethanol |

  | EtOH |

  | alcohol |

  | ethyl alcohol |

  | common solvent |

  | disinfectant |

  | flammable liquid |

  | reagent in reactions |

  | soluble in water |

Only output the converted results, do not output the conversion process!
Now process: Molecule: {question} Mol2caption answer: {answer},

**Box A.15: Prompt for generating equivalent answers to chemical caption2mol problems**

Given a textual description of a molecule, generate 10 equivalent molecular representations:
- Different SMILES strings (canonical and non-canonical)
- InChI
- Molecular formula
- Graphical structure representation (if feasible)
- Different naming conventions resolved to the same structure

Format: LaTeX \boxed{...}, 10 forms.
**Example**:    Input: "alcohol used for disinfection"
  Output:

  | C2H5OH |

  | CCO |

  | OCC |

  | InChI=1S/C2H6O/c1-2-3/h3H,2H2,1H3 |

  | ethanol |

  | EtOH |

  | ethyl alcohol |

  | common lab alcohol |

  | solvent for reactions |

Only output the converted results, do not output the conversion process!
Now process: Caption: question Predicted molecule: answer,

---

**Box A.16: Prompt for generating equivalent answers to chemical name conversion problems**

---

Given a molecule name in one format, generate 10 equivalent forms:
- IUPAC ↔ trivial/common name
- English ↔ Chinese name
- SMILES
- InChI
- Hydrate/salt forms if applicable
- Synonyms and international spelling variants

**Format**: LaTeX \boxed{...}, 10 outputs.

**Example**:

Input: Acetic acid

Output:

Acetic acid

ethanoic acid

CH3COOH

C2H4O2

Vinegar acid

Acetic acid, glacial

Acetic acid aqueous solution

Sodium acetate form

Acid CH3COOH

Only output the converted results, do not output the conversion process!

Now process: Name: {question} Predicted conversion: {answer}

---

---

**Box A.17: Prompt for generating equivalent answers to biological protein inverse folding problems**

Given the following question and an answer in protein sequence form, generate 10 different equivalent forms of this sequence, using transformations such as:
- Synonymous sequences folding into the same structure
- - Conservative amino acid substitutions (e.g., Lys ↔ Arg, Asp ↔ Glu)
- FASTA format ↔ plain sequence string
- One-letter code ↔ three-letter code
- Sequence with small tolerated mutations that preserve structure
- Adding/removing header lines in FASTA
- Using lowercase vs uppercase amino acid letters
- Introducing gap symbols to indicate alignment but preserving structure
- Representing sequence in JSON or array form
- Grouping amino acids by domains/regions but preserving the overall sequence

**Format**: Output exactly 10 forms. Each form must be wrapped in LaTeX \boxed{...}.

**Example**:
  Input: MKAILVVLLYTAA
  Output:

  $\boxed{\text{MKAILVVLLYTAA}}$

  $\boxed{\text{Met-Lys-Ala-Ile-Leu-Val-Val-Leu-Leu-Tyr-Thr-Ala-Ala}}$

  $\boxed{\text{>seq1nMKAILVVLLYTAA}}$

  $\boxed{\text{mkailvvll ytaa}}$

  $\boxed{\text{MKAILVVL–YTAA}}$

  $\boxed{\text{['M','K','A','I','L','V','V','L','L','Y','T','A','A']}}$

  $\boxed{\text{MKAILVVLLY(S/T)AA}}$

  $\boxed{\text{MKAILVVLLY RAA (Arg substitution)}}$

  $\boxed{\text{[Domain1: MKAILV, Domain2: VLLYTAA]}}$

  $\boxed{\text{MKAILVVLLYTAA (unchanged)}}$

Only output the converted results in LaTeX
boxed..., do not output the conversion process!
Now, process the following answer: Question: {question} Answer: {answer},

---

**Box A.18: Prompt for generating equivalent answers to biological protein structure prediction problems**

Given the following question and an answer in protein structure form, generate 10 different equivalent forms of this structure, using transformations such as:
- PDB format ↔ mmCIF format
- 3D atomic coordinates ↔ Cα-only coordinates
- 3D structure ↔ contact map ↔ distance matrix
- Cartesian coordinates ↔ internal torsion angles ($\phi$, $\psi$, $\chi$)
- Secondary structure representation (helix/sheet/loop) instead of full 3D
- Alternative but equivalent chain numbering or atom ordering
- Superimposed structures with RMSD < threshold
- JSON or graph-based adjacency representation of the structure
- Coarse-grained models (e.g., backbone only)
- Visual encodings (e.g., dot-bracket for secondary structure, schematic diagrams)

**Format**: Output exactly 10 forms. Each form must be wrapped in LaTeX \boxed{...}.

**Example**:

Input: PDB file with Cα coordinates of a helix

Output:

> ATOM 1 CA ALA 1 0.000 0.000 0.000

> loop helix helix sheet

> contact map: (1,5),(2,6),(3,7)

> distance matrix: [[0,1.2,...],[1.2,0,...],...]

> torsion angles $\phi = -60, \psi = -45$

> mmCIF equivalent entry

> JSON: "atoms":["res":"ALA","atom":"CA","x":0,"y":0,"z":0]

> graph: nodes=7, edges=[(1,5),(2,6),(3,7)]

> backbone-only representation

> cartoon schematic: helix symbol

Only output the converted results in LaTeX \boxed{...}, do not output the conversion process!

Now, process the following answer: Question: {question} Answer: {answer},

---

**Box A.19: Prompt for generating equivalent answers to biological transformed agent-clinic problems**

---

Given the following question and an answer in agent action/trajectory form, generate 10 different equivalent forms of this solution, using transformations such as:

- Different but equivalent action sequences leading to same outcome
- Merging multiple actions into macro-actions
- Splitting macro-actions into finer-grained steps
- JSON ↔ tabular ↔ natural language action logs
- Reordering independent actions without changing outcome
- Adding no-op (null) actions that do not affect outcome
- Representing state transitions instead of actions
- Graph or tree representation of the policy trace
- Summarizing actions at high-level clinical outcome description
- Encoding with symbolic tokens instead of natural language

**Format**: Output exactly 10 forms. Each form must be wrapped in LaTeX \boxed{...}.

**Example**:

  Input: Give drug A → Measure blood pressure → Stop treatment

  Output:

  Give drug A → Measure BP → Stop treatment

  Action1, Action2, Action3

  [Give drug A, Measure blood pressure, Stop treatment]

  MacroAction: Treat+Monitor

  State transitions: S0 → S1 → S2

  Add no-op: Give drug A → Wait → Measure BP → Stop

  JSON: "actions":["Give drug A","Measure BP","Stop"]

  Table: 1. Give drug A | 2. Measure BP | 3. Stop

  Graph representation: nodes=states, edges=actions

  Outcome: Patient stabilized after drug A

Only output the converted results in LaTeX \boxed{...}, do not output the conversion process!

Now, process the following answer: Question: question Answer: answer"""",

**Box A.20: Prompt for generating equivalent answers to biological rna structure prediction problems**

Given the following question and an answer in RNA structure form, generate 10 different equivalent forms of this structure, using transformations such as:

- Dot-bracket notation ↔ CT format ↔ BPseq format
- Base-pairing list ↔ adjacency matrix representation
- 2D secondary structure ↔ 3D atomic coordinates
- Minimal free energy structure ↔ near-optimal structure
- Adding base-pair probabilities annotation
- Grouping stems/loops/bulges into modular blocks
- JSON array representation of base pairs
- ASCII art of RNA fold representation
- Graph representation of paired/unpaired nodes
- - Annotated sequence with paired regions in brackets

**Format**: Output exactly 10 forms. Each form must be wrapped in LaTeX \boxed{...}.

**Example**:

Input: GCGCUUAGC, Structure: (((...)))

Output:

$\boxed{(((...)))}$

$\boxed{\text{dot-bracket: (((...)))}}$

$\boxed{\text{CT: 1 9, 2 8, 3 7}}$

$\boxed{\text{BPseq: 1 9, 2 8, 3 7}}$

$\boxed{\text{pairs=[(1,9),(2,8),(3,7)]}}$

$\boxed{\text{matrix: [[0,0,1,...],...]}}$

$\boxed{\text{near-optimal structure (((..).))}}$

$\boxed{\text{ASCII: stem-loop diagram}}$

$\boxed{\text{JSON: "pairs":[[1,9],[2,8],[3,7]]}}$

$\boxed{\text{annotated sequence: GCG(CUU)AGC}}$

Only output the converted results in LaTeX \boxed{...}, do not output the conversion process!

Now, process the following answer: Question: {question} Answer: {answer}

---

**Box A.21: Prompt for generating equivalent answers to biological rna inverse folding problems**

Given the following question and an answer in RNA sequence form, generate 10 different equivalent forms of this sequence, using transformations such as:

- Different sequences folding into the same target structure
- U ↔ T replacement (RNA vs DNA notation)
- FASTA format ↔ plain string sequence
- Lowercase vs uppercase bases
- Adding alignment gaps without changing fold
- JSON or array encoding of sequence
- Annotating sequence with regions (stem, loop, bulge)
- Replacing synonymous positions with alternative nucleotides that preserve folding
- Annotated sequence with secondary structure alignment marks
- Splitting long sequence into chunks and recombining

**Format**: Output exactly 10 forms. Each form must be wrapped in LaTeX \boxed{...}.

**Example**:   Input: AUGCGAU

   Output:

$\boxed{\text{AUGCGAU}}$

$\boxed{\text{augcgau}}$

$\boxed{\text{ATGCGAT}}$

$\boxed{\text{>seq1nAUGCGAU}}$

$\boxed{\text{AU-GCGAU}}$

$\boxed{\text{['A','U','G','C','G','A','U']}}$

$\boxed{\text{stem(AUG) loop(CGAU)}}$

$\boxed{\text{AUGCGAU (unchanged)}}$

$\boxed{\text{JSON: "sequence":"AUGCGAU"}}$

$\boxed{\text{chunks: AU | GC | GA | U}}$

Only output the converted results in LaTeX \boxed{...}, do not output the conversion process!

Now, process the following answer: Question: {question} Answer: {answer}

---

---

**Box A.22: Prompt for generating equivalent answers to QA problems**

You are given an answer to a general question. Generate 10 different equivalent forms of this answer, using transformations such as:
- Paraphrasing the text while keeping the meaning the same.
- Changing the sentence structure or word order.
- Using synonyms or alternative expressions.
- Expressing the answer as a list, table, or bullet points if applicable.
- Rewriting as formal statements, equations, or logical expressions if relevant.
- Approximating special constants: $\pi \approx 3.14159, e \approx 2.718$
- Angle-radian conversion: $\pi/6 = 30°$
- Dimensional conversion (e.g., $1N \rightarrow 1kg \cdot m/s^2$)

**Format**: Output exactly 10 forms. Each form must be wrapped in LaTeX \boxed{...}. Separate each answer with a newline (\n).

**Example**:
  Input: "Water boils at 100 degrees Celsius at sea level."
  Output:

  Water reaches its boiling point at 100°C at sea level.

  At sea level, the boiling temperature of water is 100 degrees Celsius.

  100°C is the temperature at which water boils at sea level.

Only output the converted results, do not output the conversion process!
Now, process the following answer: Question: {question} Answer (expression): {answer}

---

## A.2 DATA ANNOTATION AND EVALUATION

Next, we describe the configurations and prompts used during data annotation and the actual evaluation. In this process, the inputs are the question, the reference answer, and the answer to be evaluated, and the output is the correctness of the answer being evaluated. To ensure stable outputs during the experiments, a temperature of 0 is used. The prompts with CoT are shown in Box. A.23, the prompts without CoT are shown in Box. A.24, and the prompts used in the main experiments to measure prompt stability are shown in Box. A.25. The LLMs used during the data annotation process were Qwen3-30B-A3B-Instruct-2507, GPT-oss-20B, Qwen2.5-72B-Instruct, LLaMa3.3-Instruct, and CompassVerifier-32B.

---

**Box A.23: Prompt of Inference with Thinking**

As a grading reward model, your task is to evaluate whether the candidate's final answer matches the provided standard answer. You must first output a detailed step-by-step analysis, then give a final structured judgment. Do not regenerate or improve answers, only compare.

**Evaluation Protocol:**

**1. Reference Standard:**
- The standard (gold) answer is definitive and always correct.
- The question is always valid — never challenge it.
- Do not regenerate answers; only compare candidate's answer with the gold answer.

**2. Comparison Method:**
- Analyze the question's requirements and the gold answer's structure.
- Determine if the question requires exact matching or allows equivalence.
- Compare ONLY the candidate's final answer. Ignore reasoning errors.
- Ignore differences in formatting or style.
- For math expressions: check algebraic equivalence step by step; if uncertain, test numerically at multiple points.
- For multiple-choice: only compare the final choice and its content.

**3. Multi-part Answers:**
- All parts must match the gold answer exactly.
- Partial matches are incorrect.
- If not specified, order may vary. For example, $\frac{27}{7}, -\frac{8}{7}$ and $-\frac{8}{7}, \frac{27}{7}$ are equivalent.

**4. Validity Check:**
- If incomplete (cut off, unfinished sentence) → Label as **INCOMPLETE**.
- If repetitive (looping words/phrases) → Label as **REPETITIVE**.
- If explicit refusal (e.g., "I cannot answer...") → Label as **REFUSAL**.
- Gives an answer but then negates it at the end → Label as **REFUSAL**.
- Any of the above → classify as **C** with the correct error type.

**Grading Scale:**

$\boxed{A}$ **- CORRECT:**
- Matches gold exactly or equivalent (including algebraic/numeric equivalence).
- For numerical values: equivalent if equal under rounding tolerance.
- Semantic equivalence allowed.

$\boxed{B}$ **- INCORRECT:**
- Any deviation from gold.
- Partial matches for multi-part answers.

$\boxed{C}$ **- INCOMPLETE/REPETITIVE/REFUSAL:**
- Invalid answers (must specify error type).

**Execution Steps and Output:**

**Analysis step by step:**
1. First check validity (INCOMPLETE, REPETITIVE, REFUSAL).
2. Compare candidate's final answer vs. gold answer in detail.
    - Identify strict requirements (e.g., exact match, order, completeness).
    - Allow tolerances (format differences, equivalent math forms, unsimplified fractions, provide the full answer for completion-type questions).
    - Check for equivalences, e.g., $\frac{2x-7}{(x+1)(x-2)}$ and $\frac{3}{x+1} - \frac{1}{x-2}$ are equivalent.
    - Consider:
        – Factoring/expansion: $x^2 + 2x + 1 \rightarrow (x+1)^2$
        – Fraction simplification: $\frac{x^2-1}{x+1} \rightarrow x - 1$
        – Partial fraction decomposition, decimals, trig identities, substitutions, etc.
    - For multiple-choice, answer must exactly match gold or be fully equivalent.
3. Provide thorough reasoning chain, highlighting subtle equivalences or deviations.

**Final Judgment:** A/B/C

Here is your task:

<Original Question Begin>{question} <Original Question End>

<Standard Answer Begin>{gold answer} <Standard Answer End>

<Candidate's Answer Begin>{llm response} <Candidate's Answer End>

Analysis step by step (not to try solving the problem yourself) and Final Judgment:

**Box A.24: Prompt of Inference without Thinking**

As a grading reward model, your task is to evaluate whether the candidate's final answer matches the provided standard answer. You must only give a final structured judgment. Do not regenerate or improve answers, only compare.

**Evaluation Protocol:**

**1. Reference Standard:**
- The standard (gold) answer is definitive and always correct.
- The question is always valid — never challenge it.
- Do not regenerate answers; only compare candidate's answer with the gold answer.

**2. Comparison Method:**
- Analyze the question's requirements and the gold answer's structure.
- Determine if the question requires exact matching or allows equivalence.
- Compare ONLY the candidate's final answer. Ignore reasoning errors.
- Ignore differences in formatting or style.
- For math expressions: check algebraic equivalence step by step; if uncertain, test numerically at multiple points.
- For multiple-choice: only compare the final choice and its content.

**3. Multi-part Answers:**
- All parts must match the gold answer exactly.
- Partial matches are incorrect.
- If not specified, order may vary. For example, $\frac{27}{7}, -\frac{8}{7}$ and $-\frac{8}{7}, \frac{27}{7}$ are equivalent.

**4. Validity Check:**
- If incomplete (cut off, unfinished sentence) → Label as **INCOMPLETE**.
- If repetitive (looping words/phrases) → Label as **REPETITIVE**.
- If explicit refusal (e.g., "I cannot answer...") → Label as **REFUSAL**.
- Gives an answer but then negates it at the end → Label as **REFUSAL**.
- Any of the above → classify as **C** with the correct error type.

**Grading Scale:**

$\boxed{A}$ **- CORRECT:**
- Matches gold exactly or equivalent (including algebraic/numeric equivalence).
- For numerical values: equivalent if equal under rounding tolerance.
- Semantic equivalence allowed.

$\boxed{B}$ **- INCORRECT:**
- Any deviation from gold.
- Partial matches for multi-part answers.

$\boxed{C}$ **- INCOMPLETE/REPETITIVE/REFUSAL:**
- Invalid answers (must specify error type).

**Annotation Criteria:**
1. First check validity (INCOMPLETE, REPETITIVE, REFUSAL).
2. Compare candidate's final answer vs. gold answer in detail.
   - Identify strict requirements (e.g., exact match, order, completeness).
   - Allow tolerances (format differences, equivalent math forms, unsimplified fractions, provide the full answer for completion-type questions).
   - Check for equivalences, e.g., $\frac{2x-7}{(x+1)(x-2)}$ and $\frac{3}{x+1} - \frac{1}{x-2}$ are equivalent.
   - Consider:
     - Factoring/expansion: $x^2 + 2x + 1 \rightarrow (x+1)^2$
     - Fraction simplification: $\frac{x^2-1}{x+1} \rightarrow x - 1$
     - Partial fraction decomposition, decimals, trig identities, substitutions, etc.
   - For multiple-choice, answer must exactly match gold or be fully equivalent.
3. Only give the final judgment.

**Final Judgment:** A/B/C

Here is your task:

<Original Question Begin>{question} <Original Question End>

<Standard Answer Begin>{gold answer} <Standard Answer End>

<Candidate's Answer Begin>{llm response} <Candidate's Answer End>

Final Judgment:

---

**Box A.25: Prompt of Inference for the evaluation of robustness**

You are a diligent and precise assistant tasked with evaluating the correctness of responses. You will receive a question, an output sentence, and the correct answer. Your task is to determine if the output sentence accurately answers the question based on the provided correct answer. Respond with either [Correct] or [Incorrect].
- Special considerations:

1. **Multiple Answers**: If the output contains multiple answers, evaluate whether later answers modify or correct earlier ones. In such cases, compare the final answer with the correct answer. If the final answer is unclear or incorrect, respond with [Incorrect].

2. **Mathematical Problems**: If the formats differ but the answers are mathematically equivalent, respond with [Correct].

3. **Explicit Options**: If the question provides explicit candidate answers, the output will be considered correct if it clearly indicates the correct option's code or the correct option's content.

4. **No Explicit Options**: If the question does not provide explicit options, the output must align with the correct answer in content and meaning to be considered [Correct].

Question: {question}
Output sentence: {llm response}
Correct answer: {gold answer}
Judgement:

---

## A.3 DATA DETAILS

In this section, we present several data cases, focusing primarily on examples of the equivalent forms we generated. Box. A.26 shows an equivalent example in mathematics, where the answers in the Outputs will undergo both LLM annotation and human annotation. Box. A.27 presents an equivalent example in physics, Box. A.28 in chemistry, Box. A.29 in biology, and Box. A.30 provides an example for QA questions.

---

**Box A.26: A sample for generating equivalent answers to mathematical problems**

**Question**: Find all functions $f : \mathbb{Q}^+ \mapsto \mathbb{Q}^+$ such that:
$$f(x) + f(y) + 2xyf(xy) = \frac{f(xy)}{f(x+y)}.$$
**Gold Ans**: $\frac{1}{x^2}$
**Outputs:**

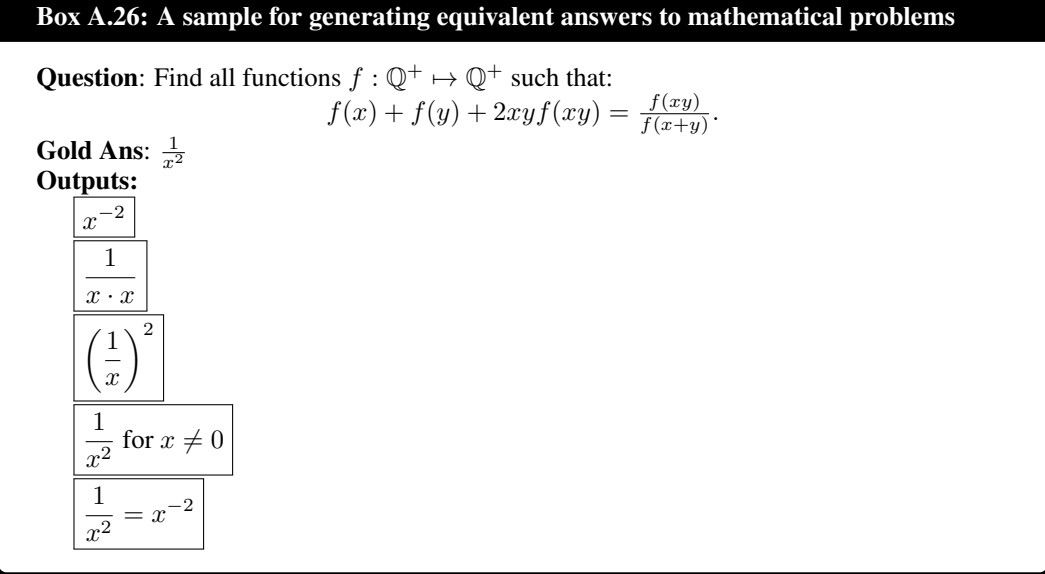

---

---

**Box A.27: A sample for generating equivalent answers to physical problems**

**Question**: Set a crystal prism with an apex angle of $60°$, and the optical axis is perpendicular to the main cross-section of the prism. The sodium yellow light is refracted in the prism in the direction of the minimum deviation angle and focused using a lens with a focal length of 1 m. What is the separation between the o-ray and e-ray spectral lines?

**Gold Ans:** 7.16 mm

**Outputs:**

7.16 mm

7.16 mm

0.00716 m

7160 $\mu$m

$7.16 \times 10^{-3}$ m

---

**Box A.28: A sample for generating equivalent answers to chemical problems**

**Question**: I'm looking for the SMILES version of this IUPAC chemical name.
N4-(3,4-difluorophenyl)-2-methyl-N6-prop-2-enylpyrimidine-4,6-diamine

**Gold Ans**: CC1=NC(=CC(=N1)NC2=CC(=C(C=C2)F)F)NCC=C

**Outputs:**

N4-(3,4-difluorophenyl)-2-methyl-N6-prop-2-enylpyrimidine-4,6-diamine

2-methyl-N6-(prop-2-en-1-yl)-N4-(3,4-difluorophenyl)pyrimidine-4,6-diamine

CC1=NC(=CC(=N1)NC2=CC(=C(C=C2)F)F)NCC=C

C12N(C=C(N1)C)C=CC(NC3=CC=C(C=C3)F)C=C2F

---

**Box A.29: A sample for generating equivalent answers to biological problems**

**Question:** Please predict the dot-bracket notation of the secondary structure directly from the RNA sequence:

UCUGAUCCAUGAUGACUUCCAAACACAUGUGGACUGUUGGAUGAAGCAUGCGGAUAUGGGACUGAGAUCAGAC

**Gold Ans:** .((((............................................))))..

**Outputs:**

.((((............................................))))..

CT: 2 30, 3 29, 4 28, 5 27, 6 26, 7 25, 8 24, 9 23, 10 22, 11 21, 12 20, 13 19, 14 18, 15 17

BPseq:
2 30, 3 29, 4 28, 5 27, 6 26, 7 25, 8 24, 9 23, 10 22, 11 21, 12 20, 13 19, 14 18, 15 17

pairs=
[(2,30),(3,29),(4,28),(5,27),(6,26),(7,25),(8,24),
(9,23),(10,22),(11,21),(12,20),(13,19),(14,18),(15,17)]

matrix:
[[0,0,0,0,0,0,0,0,0,0,0,0,0,0,0,0,0,0,0,0,0,0,0,0,0,0,0,0,0,0,0],
[0,0,0,0,0,0,0,0,0,0,0,0,0,0,0,0,0,0,0,0,0,0,0,0,0,0,0,0,0,0,1],
...,
[0,1,0,0,0,0,0,0,0,0,0,0,0,0,0,0,0,0,0,0,0,0,0,0,0,0,0,0,0,0,0]]

---

**Box A.30: A sample for generating equivalent answers to QA problems**

**Question**: What's the name of the active president of the United States in Season 1 of Motherland: Fort Salem?
**Gold Ans**: President Kelly Wade
**Outputs:**

The active president of the United States in Season 1 of Motherland: Fort Salem is named Kelly Wade.

Kelly Wade is U.S. President in Season 1 of Motherland: Fort Salem.

In the first season of Motherland: Fort Salem, the president is Kelly Wade.

Kelly Wade is the incumbent U.S. leader in Season 1 of Motherland: Fort Salem.

In Season 1 of Motherland: Fort Salem, Kelly Wade serves as the U.S. president.

---

# B   DETAILS OF TRAINING

In this section, we present the parameter configurations used during the training process. The training mainly involves two parts: SFT and RL. For SFT, we adopt full fine-tuning, and the detailed parameter configurations are shown in Tab. 6. For RL, a modified version of GRPO is used, with detailed parameters also provided in Tab. 7.

## C    LIMITATIONS AND FUTURE WORK

In this section, we discuss the limitations of our work and outline directions for future research. We propose a verifier for scientific verification tasks that demonstrates strong reasoning capabilities, achieving high performance with concise and interpretable reasoning outputs. However, some scenarios demand both high accuracy and extreme efficiency. In future work, we plan to leverage the model's explicit reasoning abilities to further enhance its implicit reasoning, allowing it to maintain strong performance even without explicitly generating detailed reasoning steps. This approach could provide significant efficiency gains while preserving the model's reliability and robustness across a wider range of scientific verification tasks.

Table 6: SFT Configuations for SCI-Verifier.

| Parameter | Value |
| --- | --- |
| BF16 | True |
| Gradient Checkpointing | False |
| Learning Rate | $5 \times 10^{-5}$ |
| LR Scheduler Type | cosine_with_min_lr |
| Minimum LR Rate | 0.1 |
| Packing | False |
| Maximum Sequence Length | 1024 |
| Maximum Steps | -1 |
| Number of Training Epochs | 2 |
| Per Device Train Batch Size | 2 |
| Per Device Eval Batch Size | 16 |
| GPUs Per Node | 4 |
| Number of Nodes | 1 |
| Seed | 42 |
| Use Liger Kernel | True |
| Warmup Ratio | 0.02 |

Table 7: RL Configuations for SCI-Verifier.

| Parameter | Value |
| --- | --- |
| BF16 | True |
| Temperature | 1.0 |
| Top p | 1.0 |
| Clip Ratio Low | 0.2 |
| Clip Ratio High | 0.28 |
| Max Response Length | 2048 |
| Overlong Buffer Len | 1024 |
| Learning Rate | $1 \times 10^{-6}$ |
| Number of Training Epochs | 20 |
| GPUs Per Node | 4 |
| Number of Nodes | 1 |
| Seed | 42 |

## D    THE USE OF LARGE LANGUAGE MODELS

In our paper, LLMs are first used to polish the writing, improving the clarity and readability of the manuscript. At the same time, as mentioned multiple times in the main text and Appendix, we employ LLMs to generate and annotate training and test data. Since LLM outputs can sometimes be unreliable, as noted in the text, all selected data are subsequently manually re-annotated by human experts and carefully filtered.

# E    DETAILS OF SCI-VERIFIER

## E.1    CASES OF SCI-VERIFIER

In this part, we present representative cases showing how SCI-Verifier evaluates model outputs when Chain-of-Thought (CoT) is enabled, making its reasoning trajectory more transparent. We select one example from each domain, and the detailed step-by-step results are provided in Box E.1 to Box E.5. In this setting, a verifier output of A indicates that it judges the answer to be correct, while an output of B indicates that it judges the answer to be incorrect.

---

**Box E.1: A sample for SCI-Verifier generating answers to Math problems**

**Question**: Let $ABCD$ be a quadrilateral with $A = (3,4), B = (9,-40), C = (-5,-12), D = (-7,24)$. Let $P$ be a point in the plane (not necessarily inside the quadrilateral). Find the minimum possible value of $AP + BP + CP + DP$.

**Gold Ans**: $16\sqrt{17} + 8\sqrt{5}$

**LLM Ans (extracted)**: $66 + 8\sqrt{5}$

**Gold Judgment**: False

**Verifier Judgment:** To evaluate whether the candidate's final answer matches the provided standard answer, we will follow the evaluation protocol step by step.

Analysis Step by Step

1. Check Validity: The candidate's answer is complete and does not show any signs of repetition or refusal to answer. Therefore, we can proceed to compare the final answer directly.

2. Compare Candidate's Final Answer vs. Gold Answer: - The candidate's final answer is $(66 + 8\sqrt{5})$. - The gold answer is $(16\sqrt{17} + 8\sqrt{5})$.

We need to determine if these two expressions are equivalent or if there is a significant difference. Let's break down the components: - Both answers contain the term $(8\sqrt{5})$, which is common.

- The remaining terms are $(66)$ and $(16\sqrt{17})$.

To check for equivalence, we can compare the numerical values of these terms: - $(66)$ is a constant. - $(16\sqrt{17})$ is approximately $(16 \times 4.123 = 65.968)$.

Clearly, $(66)$ and $(65.968)$ are very close in value, but they are not exactly the same. The difference is minimal, but it is still present.

3. Provide a Thorough Reasoning Chain: - The candidate's answer $(66 + 8\sqrt{5})$ is very close to the gold answer $(16\sqrt{17} + 8\sqrt{5})$, but not exactly the same. - The difference between $(66)$ and $(16\sqrt{17})$ is approximately $(0.032)$, which is negligible in most practical contexts but not zero.

Final Judgment:

Given that the candidate's final answer $(66 + 8\sqrt{5})$ is very close to the gold answer $(16\sqrt{17} + 8\sqrt{5})$ but not exactly the same, we can conclude that the candidate's answer is incorrect.

Therefore, the final judgment is: [ boxed{B} ].

---

---

**Box E.2: A sample for SCI-Verifier generating answers to Physics problems**

**Question**: There are two identical objects, with internal energy given by $\mathcal{U} = NCT$, where $N$ is the number of particles in the object and $C$ is a constant. Initially, the temperatures of the two objects are $T_1$ and $T_2$, respectively. Now, the two objects act as high and low temperature heat sources to drive a Carnot engine, and finally, the two objects reach a common temperature $T_f$. Find the work $W$ done by the Carnot engine.

**Gold Ans**: $NC\left(T_1 + T_2 - 2\sqrt{T_1 T_2}\right)$

**LLM Ans (extracted)**: $NC\left(\sqrt{T_1} - \sqrt{T_2}\right)^2$

**Gold Judgment**: True

**Verifier Judgment:** To evaluate whether the candidate's final answer matches the provided standard answer, we will follow the evaluation protocol step by step.

Analysis Step by Step

1. Check Validity: The candidate's answer is complete and does not show any signs of repetition or refusal to answer. Therefore, we can proceed to compare the final answer with the gold answer.

2. Compare Candidate's Final Answer vs. Gold Answer:

- The candidate's final answer is $NC\left(\sqrt{T_1} - \sqrt{T_2}\right)^2$.

- The gold answer is $NC\left(T_1 + T_2 - 2\sqrt{T_1 T_2}\right)$

We need to determine if these two expressions are equivalent. Let's expand the candidate's answer:

$NC\left(\sqrt{T_1} - \sqrt{T_2}\right)^2 = NC\left(T_1 - 2\sqrt{T_1 T_2} + T_2\right)$ This can be rewritten as:

$NC\left(T_1 + T_2 - 2\sqrt{T_1 T_2}\right)$

This is exactly the same as the gold answer.

3. Provide a Thorough Reasoning Chain:

- The candidate's answer $NC\left(\sqrt{T_1} - \sqrt{T_2}\right)^2$ simplifies to $NC\left(T_1 + T_2 - 2\sqrt{T_1 T_2}\right)$, which is identical to the gold answer $NC\left(T_1 + T_2 - 2\sqrt{T_1 T_2}\right)$.

Final Judgment

Since the candidate's final answer matches the gold answer exactly, the final judgment is: [boxed{A}],

---

**Box E.3: A sample for SCI-Verifier generating answers to Chemistry problems**

**Question**: How would you express this IUPAC name in SMILES format?

**Gold Ans**: The SMILES representation for 2-diazanylidene-4-methylpentan-1-imine is CC(C)CC(=NN)C=N

**LLM Ans (extracted)**: CC(C)CC(=NN)C=N

**Gold Judgment**: True

**Verifier Judgment:** To evaluate whether the candidate's final answer matches the provided standard answer, we will follow the evaluation protocol step by step.

1. First check validity: The candidate's answer is complete and does not show any signs of repetition or refusal. Therefore, we can proceed to compare the candidate's final answer with the gold answer.

2. Compare candidate's final answer vs. gold answer:

- The candidate's final answer is 'CC(C)CC(=NN)C=N'.

- The gold answer is also 'CC(C)CC(=NN)C=N'.

Since the candidate's final answer exactly matches the gold answer, there are no deviations or equivalences that need to be considered. The structure and content of the SMILES notation are identical.

3. Provide a thorough reasoning chain: The candidate's answer and the gold answer are identical, indicating that the candidate has correctly expressed the IUPAC name in SMILES format.

Final Judgment:

boxed{A} - CORRECT: The candidate's final answer matches the gold answer exactly.

---

**Box E.4: A sample for SCI-Verifier generating answers to Biology problems**

**Question**: Please predict the dot-bracket notation of the secondary structure directly from the RNA sequence: AUCCUUAAAACAUGU-UCGCUGGCGUAGCUGGCAGCAUCCCAGGCAGUCAAGGCCAUCAA-GACUGCGGCGAACCUACAGUUCGCUGGCUGAUCUCAUCUCCGGACGCUCAUCGUCC AUUCCAACCCCAAGGGACCAUUCAUCCCUAUGGGGGUGA.

**Gold Ans**: .........(.(((..(((((((...))))))).))).....(((((...........)))))(((((.....)))))))............................ .(.(....(((((((((.........)))))))).))

**LLM Ans (extracted)**: dot-bracket: .........(.(((..(((((((...))))))).))).....(((((...........)))))(((((.....))))) ))..............................(.(....(((((((((.........)))))))).))..(.(((.....))))....(((((.......))))).................

**Gold Judgment**: False

**Verifier Judgment:** To evaluate the candidate's final answer against the standard answer, we will follow the evaluation protocol step by step.
1. First check validity (INCOMPLETE, REPETITIVE, REFUSAL). The candidate's answer is complete and not repetitive or refusing to answer.
2. Compare candidate's final answer vs. gold answer in detail:
- The candidate's answer is: '.........(.(((..(((((((...))))))).))).....(((((...........)))))(((((.....))))))..... ..........................(.(....(((((((((.........)))))))).))..(.(((.....))))....(((((.......)))))...............'
- The gold answer is: '.........(.(((..(((((((...))))))).))).....(((((...........)))))(((((.....)))))))............... ................(.(....(((((((((.........)))))))).))'
We can see that the candidate's answer has an additional part at the end: '..(.(((.....))))....(((((.......))))))...............'. This part is not present in the gold answer.
3. Provide a thorough reasoning chain, highlighting subtle equivalences or deviations. The candidate's answer includes an extra section that is not part of the gold answer. This deviation means that the candidate's answer does not match the gold answer exactly.
Final Judgment: boxed{B} - INCORRECT, because the candidate's final answer includes an additional part that is not present in the gold answer.

---

**Box E.5: A sample for SCI-Verifier generating answers to Scientific QA problems**

**Question**: A packet filter firewall filters at the.
**Gold Ans**: Network or transport layer
**LLM Ans (extracted)**: network layer
**Gold Judgment**: True
**Verifier Judgment:** To evaluate whether the candidate's final answer matches the provided standard answer, we will follow the evaluation protocol outlined:
1. First check validity: The candidate's answer is not incomplete, repetitive, or a refusal. It is a clear and concise response.
2. Compare candidate's final answer vs. gold answer in detail:
- The candidate's answer is network layer.
- The gold answer is Network or transport layer.
- The candidate's answer is very close to the gold answer. The only difference is the use of a backslash before layer in the candidate's answer, which is likely a typographical error. The intended meaning is the same as the gold answer, which refers to the network layer or the transport layer.
3. Provide a thorough reasoning chain, highlighting subtle equivalences or deviations:
- The candidate's answer network layer can be interpreted as referring to the network layer, which is one of the layers in the OSI model. The gold answer Network or transport layer indicates that either the network layer or the transport layer could be correct. Since the candidate's answer is essentially referring to the network layer, it aligns with the gold answer.
Final Judgment:
Given that the candidate's answer is essentially correct, with only a minor typographical error, and considering the context of the question, the final judgment is:
boxed{A} - CORRECT.",

## E.2 Failure Cases of SCI-Verifier

In this section, we analyze several failure cases of SCI-Verifier to better understand its current limitations and guide future improvements. We find that most errors arise during the equivalence-checking phase, where SCI-Verifier makes incorrect reasoning steps when deciding whether two expressions are equivalent. While such mistakes do not always lead to an incorrect final verdict, they reveal a clear weakness of the current SCI-Verifier.

In Box E.6 and Box E.7, we present several representative failure cases from SCI-Verifier-8B, with the erroneous reasoning steps highlighted in red. In this setting, a verifier output of A indicates that it judges the answer to be correct, while an output of B indicates that it judges the answer to be incorrect. These errors fall into two categories: incorrect reasoning with a correct final answer, and incorrect reasoning that leads to an incorrect final answer. These examples show that, despite our efforts to enhance SCI-Verifier's own reasoning capability, it remains imperfect and can still produce flawed intermediate reasoning. This may be related to the capability of the base model. In future versions, we plan to explicitly target and mitigate these issues to further improve the robustness and reliability of SCI-Verifier.

---

**Box E.6: A case where SCI-Verifier reasons wrongly but judges correctly.**

**Question**: How many different collections of 9 letters are there? A letter can appear multiple times in a collection. Two collections are equal if each letter appears the same number of times in both collections.
**Gold Ans**: $\binom{34}{9}$
**LLM Ans (extracted)**: 2517660288
**Gold Judgment**: False
**Verifier Judgment:** To evaluate whether the candidate's final answer matches the provided standard answer, we will follow the evaluation protocol step by step.
Step 1: Check Validity
- The candidate's answer is 2517660288.
- There are no signs of incompleteness, repetition, or refusal in the candidate's response.
- Therefore, we proceed to compare the candidate's final answer with the gold answer.
Step 2: Compare Candidate's Final Answer vs. Gold Answer
- The gold answer is ($\binom{34}{9}$).
- The candidate's answer is 2517660288.

To determine if these two values are equivalent, we need to calculate ($\binom{34}{9}$) and compare it to 2517660288.
$\binom{34}{9} = \frac{34!}{9!(34-9)!} = \frac{34!}{9! \cdot 25!}$
Calculating this directly can be complex, but we can use a calculator or software to find the value: $\textcolor{red}{\binom{34}{9} = 5,311,735}$
However, the candidate's answer is 2517660288, which is significantly larger than 5,311,735. This indicates that the candidate's answer does not match the gold answer.
Step 3: Provide a Thorough Reasoning Chain
- The candidate's answer 2517660288 is not equal to $\binom{34}{9}$, which is 5,311,735.
- There are no other forms of equivalence (such as different representations of the same number) that would make the candidate's answer correct.
Final Judgment
Since the candidate's final answer does not match the gold answer, and there are no other forms of equivalence, the final judgment is: [boxed{B}].

---

---

**Box E.7: A case where SCI-Verifier reasons wrongly and judges wrongly.**

**Question**: Find all pairs of positive integers $(x, y)$ with the following property:If $a, b$ are relative prime and positive divisors of $x^3 + y^3$, then $a + b - 1$ is divisor of $x^3 + y^3$.(Cyprus)

**Gold Ans**:$(2^k, 2^k), (2 \cdot 3^k, 3^k), (3^k, 2 \cdot 3^k)$ for non-negative integers $k$.

**LLM Ans (extracted)**: (2,2)

**Gold Judgment**: False

**Verifier Judgment:** To evaluate the candidate's final answer against the standard answer, we will follow the evaluation protocol step by step.

1. Check Validity: The candidate's answer is complete and does not show any signs of repetition or refusal. Therefore, we can proceed to the comparison.

2. Comparison Method:

- The question asks for all pairs of positive integers(x, y) such that if (a, b) are relative prime and positive divisors of $(x^3 + y^3)$, then (a + b - 1) is a divisor of $(x^3 + y^3)$.

- The candidate's final answer is (2, 2). We need to check if this pair satisfies the condition given in the question.

3. Detailed Comparison:

- For the pair (2, 2), we have (x = 2) and (y = 2). Therefore, $(x^3 + y^3 = 2^3 + 2^3 = 8 + 8 = 16)$.

- The divisors of 16 are(1, 2, 4, 8, 16). Since (a) and (b) are relative primes, the only pairs of divisors that are relative primes are((1, 2)), ((1, 4)),((1, 8)), ((1, 16)), (2, 4), (2, 8), ((2, 16)), ((4, 8)), ((4, 16)), and ((8, 16)). However, since (a) and (b) must be relative primes, we only consider ((1, 2)), ((1, 4)), ((1, 8)), and ((1, 16)).

- For each pair, we check if (a + b - 1) is a divisor of 16:

- For ((1, 2)), (a + b - 1 = 1 + 2 - 1 = 2), and 2 is a divisor of 16.

- For ((1, 4)), (a + b - 1 = 1 + 4 - 1 = 4), and 4 is a divisor of 16.

- For ((1, 8)), (a + b - 1 = 1 + 8 - 1 = 8), and 8 is a divisor of 16.

- For ((1, 16)), (a + b - 1 = 1 + 16 - 1 = 16), and 16 is a divisor of 16.

- All these conditions are satisfied, so the pair ((2, 2)) meets the criteria.

4. Final Judgment:

- The candidate's final answer ((2, 2)) is correct according to the standard answer, which includes $((2^k, 2^k))$ for non-negative integers (k). Since (k = 0) gives us $((2^0, 2^0) = (1, 1))$ and (k = 1) gives us $((2^1, 2^1) = (2, 2))$, the candidate's answer is indeed part of the standard answer.

Therefore, the final judgment is: boxed{A}

