# OpenReview forum: "SCI-Verifier: Scientific Verifier with Thinking"
_ICLR.cc/2026/Conference — ICLR 2026 Poster_

### Official Review · Reviewer_Hp78 · 2025-10-22

**Soundness:** 3
**Presentation:** 3
**Contribution:** 3
**Rating:** 8
**Confidence:** 4

**Summary:**

This paper introduces a challenging benchmark for scientific verification that spans five distinct subjects and domains. In addition, the authors present a reasoning-based scientific verifier trained on carefully curated datasets, which substantially outperforms existing verification models. Interestingly, they highlight the inherent difficulty of equivalence judgment—an essential aspect of scientific evaluation. This work is well-written and solid.

**Strengths:**

1. The writing is clear and easy to follow.

2. The paper highlights the importance of equivalence transformations in answers, identifying them as a crucial and challenging aspect of verification. Once this issue is addressed, the results show remarkable improvement.

3. This work is comprehensive and well-executed, covering data creation, benchmark construction, and verifier training—each of which involves substantial effort—and it achieves outstanding performance overall.

**Weaknesses:**

1. There is a lack of error analysis for Sci-Verifier, making it difficult to understand what problems still remain.

2. There is a lack of human evaluation. Although Sci-Verifier performs well on the benchmark, it is unclear how consistent its results are with human judgments.

**Questions:**

1. Could you conduct an error analysis on the problems that the current verifier fails to handle well? I’d like to understand what major issues still exist in the current verification process.

2. Could you sample some data and conduct a comparative analysis between human evaluations and the verifier’s results?

---

> ### Author Response · Authors · 2025-11-21
> **Official Comment by Authors**
>
> **Weakness1&Question1.There is a lack of error analysis for Sci-Verifier.**
>
> A1. Thank you for raising this question. We provide a detailed error analysis in Appendix E.2.  In addition, we present below examples of two types of remaining errors in SCI-Verifier: cases where the reasoning is flawed but the final answer is still correct, and cases where incorrect reasoning leads to an incorrect answer. A detailed analysis is provided in Appendix E.2. In this setting, a verifier output of A indicates that it judges the answer to be correct, while an output of B indicates that it judges the answer to be incorrect.
>
> ```LaTeX
> A case where SCI-Verifier reasons wrongly but judges correctly.
>
> \textbf{Question}:
> How many different collections of 9 letters are there? A letter can appear multiple times in a collection. Two collections are equal if each letter appears the same number of times in both collections.
>
> \textbf{Gold Ans}:$\binom{34}{9}$
>
> \textbf{LLM Ans  (extracted)}: 2517660288
>
> \textbf{Gold Judgment}: False
>
> \textbf{Verifier Judgment:} To evaluate whether the candidate's final answer matches the provided standard answer, we will follow the evaluation protocol step by step.
>
> Step 1: Check Validity
>
> - The candidate's answer is \"2517660288\".
>
> - There are no signs of incompleteness, repetition, or refusal in the candidate's response.
>
> - Therefore, we proceed to compare the candidate's final answer with the gold answer.
>
> Step 2: Compare Candidate’s Final Answer vs. Gold Answer
>
> - The gold answer is ($\binom{34}{9}
> $).
>
> - The candidate's answer is \"2517660288\".
>
> To determine if these two values are equivalent, we need to calculate ($\binom{34}{9}
> $) and compare it to 2517660288.
>
> $\binom{34}{9}
> = \frac{34!}{9!(34-9)!} = \frac{34!}{9! \cdot 25!}$
>
> Calculating this directly can be complex, but we can use a calculator or software to find the value:{\textcolor{red}{$\binom{34}{9}
> = 5,311,735$}}
>
> However, the candidate's answer is 2517660288, which is significantly larger than 5,311,735. This indicates that the candidate's answer does not match the gold answer.
>
> Step 3: Provide a Thorough Reasoning Chain
>
> - The candidate's answer 2517660288 is not equal to $\binom{34}{9}$, which is 5,311,735.
>
> - There are no other forms of equivalence (such as different representations of the same number) that would make the candidate's answer correct.
>
> Final Judgment
>
> Since the candidate's final answer does not match the gold answer, and there are no other forms of equivalence, the final judgment is:
> [boxed\{B\}].
> ```
> ```LaTeX
> A case where SCI-Verifier reasons wrongly and judges wrongly.
>
> \textbf{Question}:
> Find all pairs of positive integers $(x,y)$ with the following property:If $a,b$ are relative prime and positive divisors of   $ x^3 + y^3$, then  $a+b - 1$ is divisor of $x^3+y^3$.(Cyprus)
>
> \textbf{Gold Ans}:$(2^k, 2^k), (2 \cdot 3^k, 3^k), (3^k, 2 \cdot 3^k) \text{ for non-negative integers } k.$
>
> \textbf{LLM Ans  (extracted)}: (2,2)
>
> \textbf{Gold Judgment}: False
>
> \textbf{Verifier Judgment:} To evaluate the candidate's final answer against the standard answer, we will follow the evaluation protocol step by step.
>
> 1. Check Validity: The candidate's answer is complete and does not show any signs of repetition or refusal. Therefore, we can proceed to the comparison.
>
> 2. Comparison Method:
>
> - The question asks for all pairs of positive integers(x, y) such that if (a, b) are relative prime and positive divisors of
> $(x^3 + y^3)$, then (a + b - 1) is a divisor of $(x^3 + y^3)$.
>
> - The candidate's final answer is (2, 2). We need to check if this pair satisfies the condition given in the question.
>
> 3. Detailed Comparison:
>
> - For the pair (2, 2), we have (x = 2) and (y = 2). Therefore, $(x^3 + y^3 = 2^3 + 2^3 = 8 + 8 = 16)$.
>
> - The divisors of 16 are(1, 2, 4, 8, 16). Since (a) and (b) are relative primes, the only pairs of divisors that are relative primes are((1, 2)), ((1, 4)),((1, 8)), ((1, 16)), (2, 4), (2, 8), ((2, 16)), ((4, 8)), ((4, 16)), and ((8, 16)). However, since (a) and (b) must be relative primes, we only consider ((1, 2)), ((1, 4)), ((1, 8)), and ((1, 16)).
>
> - For each pair, we check if (a + b - 1) is a divisor of 16:
>
> - For ((1, 2)), (a + b - 1 = 1 + 2 - 1 = 2), and 2 is a divisor of 16.
>
> - For ((1, 4)), (a + b - 1 = 1 + 4 - 1 = 4), and 4 is a divisor of 16.
>
> - For ((1, 8)), (a + b - 1 = 1 + 8 - 1 = 8), and 8 is a divisor of 16.
>
> - For ((1, 16)), (a + b - 1 = 1 + 16 - 1 = 16), and 16 is a divisor of 16.
>
> - All these conditions are satisfied, so the pair ((2, 2)) meets the criteria.
>
> 4. Final Judgment:
>
> - The candidate's final answer ((2, 2)) is correct according to the standard answer, which includes $((2^k, 2^k))$ for non-negative integers (k). Since (k = 0) gives us $((2^0, 2^0) = (1, 1))$ and (k = 1) gives us $((2^1, 2^1) = (2, 2))$, the candidate's answer is indeed part of the standard answer.
>
> Therefore, the final judgment is:  boxed\{A\}
> ```

---

> > ### Author Response · Authors · 2025-11-21
> > **Official Comment by Authors**
> >
> > **Weakness2&Question2. There is a lack of human evaluation. Although Sci-Verifier performs well on the benchmark, it is unclear how consistent its results are with human judgments.**
> >
> > A2. Thank you for raising this question. Note that all data in our proposed SCI-VerifyBench are human-annotated, so any verifier error on SCI-VerifyBench directly corresponds to a disagreement with human labels. The analysis of discrepancies between SCI-Verifier and human judgments is provided in our response A1. We find that most errors arise during the equivalence-checking phase, where SCI-Verifier makes incorrect reasoning steps when deciding whether two expressions are equivalent. While such mistakes do not always lead to an incorrect final verdict, they reveal a clear weakness of the current SCI-Verifier. In future versions, we plan to explicitly target and mitigate these issues to further improve the robustness and reliability of SCI-Verifier.

---

> > > ### Comment · Reviewer_Hp78 · 2025-11-25
> > > **Thank you**
> > >
> > > Thank you for the detailed response. Your clarifications adequately address my concerns, and I appreciate the additional explanations and evidence provided. I have no further questions.

---

> ### Author Response · Authors · 2025-11-26
> **Official Comment by Authors**
>
> Dear Reviewer,
>
> Thank you very much for your thoughtful follow-up and for carefully considering our response. We sincerely appreciate your time and constructive feedback.

---

### Official Review · Reviewer_x8GA · 2025-10-26

**Soundness:** 2
**Presentation:** 3
**Contribution:** 3
**Rating:** 4
**Confidence:** 4

**Summary:**

This paper introduces (i) SCI‑VerifyBench, a cross‑disciplinary verification benchmark built from real LLM responses and domain‑specific equivalence transformations across Math, Physics, Chemistry, Biology, and general scientific QA, and (ii) SCI‑Verifier, a reasoning‑augmented verifier trained with SFT+RL that aims for concise, stable judgments. The test set comprises 2500 human‑verified items (500 per domain) drawn from 15k QA pairs and 100k+ LLM responses; the construction explicitly includes equivalence‑transformed answers to stress verifiers. On SCI‑VerifyBench, an 8B SCI‑Verifier matches or exceeds closed‑source models (e.g., near GPT‑5), and it also performs strongly on VerifierBench and VerifyBench‑Hard.

**Strengths:**

- The motivation is good. The paper articulates why scientific verification is challenging (multi‑step reasoning, many equivalent forms...) and positions verification as foundational to evaluation and reliability. The narrative is easy to follow.

- The scope is reasonable. SCI‑VerifyBench spans 5 disciplines, filters tasks, and injects equivalence transformations (e.g., unit/name conversions, algebraic rewrites, symbolic identities) to create realistic, hard cases, with a 2500‑item test set that requires unanimous human agreement (but I do have some concerns on this benchmark).

- The authors add CoT to boost accuracy across models; the paper highlights this with Fig.1 (and ablations in Fig.5), and builds SCI‑Verifier around short CoT distilled via SFT and then refined with DAPO/GRPO‑style RL.

- The performance seems promising. SCI‑Verifier‑8B achieves good performance on SCI‑VerifyBench compared with other models like GPT5. On equivalence‑only subsets, many LLMs—including GPT‑5—drop sharply (below 50% in Math/Physics), while SCI‑Verifier remains substantially higher. The authors also show that SCI‑Verifier is less sensitive to prompt changes than general models. Pls report statistical significance or CIs around headline numbers.

**Weaknesses:**

(also questions for authors)

- The pipeline generates 100k+ responses using eight models, and five LLMs assist in judging equivalences; only 2500 high‑disagreement samples (500/domain) are escalated to human annotation for the test set, while much of the training set relies on LLM labels (≈14k after filtering). Although test items require full human agreement, the selection by model disagreement may be biased toward tricky/ambiguous patterns, and the training labels inherit LLM biases.  Besides, regarding the annotation, please provide inter‑annotator agreement (IAA) or adjudication statistics are reported for the human‑labeled set (beyond the “full agreement” requirement) to gauge label consistency. The benchmark & data generation part is the core contribution in this paper, so this part is very important and must be carefully evaluated and justified.

- What counts as “equivalent” is sometimes approximate, not strictly equal. Several transformation templates explicitly include approximations (e.g., Taylor expansions or “approx. 80°C”, descriptive lab phrases like “reflux”) and semantic paraphrases in QA. These are context‑dependent rather than strictly equivalent and could blur the task definition or inflate ambiguity.

- Method novelty is limited. SCI‑Verifier uses a fairly standard two‑stage SFT + RL recipe with short‑CoT distillation and a DAPO/GRPO‑style objective plus a length penalty—sensible engineering, but incremental. The paper’s main novelty is task focus + dataset rather than a new learning principle.

- Because the equivalence categories used to synthesize data appear in both *training and test pipelines*, there is a risk that models overfit to those templates rather than general principles of equivalence (?). The authors partly mitigate this with human curation for the test set, but a strict “held‑out transformation family” evaluation would strengthen the generalization claim.

**Questions:**

- Human annotation quality. As I mentioned above, please report IAA and adjudication rates for the 2,500 test items; also, clarify annotator backgrounds and per‑domain expertise distributions in detail...

- How do you formally distinguish strict equivalence from approximate/semantic equivalence? Would you consider splitting SCI‑VerifyBench into strict vs approximate tracks to avoid conflating them?

- Could specialized verifiers be run with a CoT‑enabled prompt (when feasible) to check whether the gap persists?

- Maybe I missed something. Do you have results where entire transformation families seen in training are held out in testing?

- Please report wall‑clock latency and cost/token comparisons vs GPT‑5 and CompassVerifier on SCI‑VerifyBench.

**Details Of Ethics Concerns:**

/

---

> ### Author Response · Authors · 2025-11-21
> **Official Comment by Authors**
>
> **Weakness1. The selection by model disagreement may be biased toward tricky/ambiguous patterns, and the training labels inherit LLM biases.**
>
> A1. Thank you for your comments on our benchmark construction. Your concern has two parts, which we address separately.
>
> First, regarding the disagreement between LLMs and between LLMs and human experts: we agree this makes the dataset more challenging. This is intentional. Our goal is to evaluate verifier accuracy under *difficult* conditions, which is crucial for complex scientific domains and for training/evaluating LLMs targeting top-tier tasks in math/physics and other sciences. In such settings, answers are much more complex in form (e.g., IMO-level solutions vs. GSM8K), so verifiers must be tested in correspondingly complex environments. Simpler settings are already covered by benchmarks such as VerifyBench, on which we also report results in Table 4 of the original manuscript.
>
> Second, for the training set, a large portion of labels come from cases where multiple LLMs fully agree. Given the scale of the training data, full manual labeling is infeasible, and using consensus among LLMs is how we ensure annotation quality at scale. On the one hand, it is worth noting that the training data does not exhibit any preference or bias toward harder examples. This is mainly evidenced by the superior performance of our trained model on VerifyBench and VerifierBench in Table 4 of the paper, indicating that it has not overfit to the equivalence-transformation data. Moreover, only about 10% of the training set consists of difficult examples, while the remaining 90% are cases on which multiple LLMs reach consensus, which makes overfitting unlikely. On the other hand, introducing the LLM’s bias is unavoidable. However, based on manual spot-checking, when the LLM’s judgments are consistent, the accuracy exceeds 95%. Thus, this bias is acceptable, and we ultimately adopted this solution. For these reasons, we believe our dataset construction is reasonable and well-suited to evaluating verifier capability in challenging regimes.
>
> **Weakness1&Question1.Provide inter‑annotator agreement (IAA) or adjudication statistics are reported for the human‑labeled set. Also, clarify annotator backgrounds and per‑domain expertise distributions in detail.**
>
> A2. Thank you for this question. The equivalence forms used for our IAA study and those described in Appendix A.1 are essentially the same: all formats listed in the prompt are treated as equivalent during human evaluation. For responses that annotators find uncertain or ambiguous, we label them as *uncertain*. To ensure label quality, samples that annotators judged as uncertain were not treated as true and were excluded from the final test set.
>
> In total, we annotated 5,000 examples by human experts, including 2,500 with equivalence transformations. Each example was independently labeled by at least two annotators. For 51 cases where the two annotators disagreed, a third annotator was involved, and we then selected 2,500 examples as the final test set, balancing true/false labels and subject areas.
>
> All primary annotators hold at least a bachelor’s degree, many hold PhDs in computer science, mathematics, or physics, and the annotator team includes multiple PhD holders(>5) in physics, mathematics, chemistry, and biology.

---

> > ### Author Response · Authors · 2025-11-21
> > **Official Comment by Authors**
> >
> > **Weakness2&Question2. What counts as “equivalent” is sometimes approximate, not strictly equal. How do you formally distinguish strict equivalence from approximate/semantic equivalence? Would you consider splitting SCI‑VerifyBench into strict vs approximate tracks to avoid conflating them?**
> >
> > A3. We thank the reviewer for this insightful question and agree that introducing equivalence notions inevitably involves some degree of approximation. However, such equivalence is pervasive in how correctness is judged across essentially all scientific disciplines: for example, trigonometric identities in mathematics, tolerable decimal rounding in physics, or equivalent molecular formulas in chemistry and biology. These forms of equivalence are routinely accepted not only in exam grading[1,2], but also in state-of-the-art scientific derivations and published results[3].
> >
> > To control the approximation introduced by equivalence in our benchmarks, we proceed differently for evaluation and training. For the dev/test sets, multiple human experts label each example with access to the question requirements, and any approximation that does *not* meet the stated requirements (e.g., an overly large rounding error) is explicitly marked as incorrect. Our evaluation strictly follows these human labels, which effectively filters out unacceptable approximate equivalences.
> >
> > For the training set, fully manual annotation is infeasible due to scale. Instead, we retain only those examples on which multiple LLMs agree and use this consensus as training supervision. This inevitably introduces a small amount of approximate-equivalence noise (cases that a strict human grader might reject), but in practice this behaves as reasonable training noise rather than harmful bias. Empirically, our model’s leading performance on both VerifierBench and VerifyBench-hard in Table 4 demonstrates that this design does not degrade, and in fact supports, robust generalization. Moreover, in a human audit of a sampled subset of the training data, we found that when LLMs reach consensus, the label accuracy exceeds 95%, indicating that the residual noise level is well within an acceptable range.
> >
> > In summary, incorporating equivalence-based data does not cause severe overfitting or undermine rigor; on the contrary, it is necessary for a verifier operating in scientific domains, where well-controlled forms of equivalence are an integral part of how correct reasoning and answers are defined.

---

> > > ### Author Response · Authors · 2025-11-21
> > > **Official Comment by Authors**
> > >
> > > **Weakness3.Method novelty is limited. SCI‑Verifier uses a fairly standard two‑stage SFT + RL recipe—sensible engineering, but incremental. The paper’s main novelty is task focus + dataset rather than a new learning principle.**
> > >
> > > A4. Thank you for raising this point. The main contributions of our work are: (1) We identify and formalize the verification bottleneck for training and evaluation of LLMs in scientific domains. (2) We show that CoT can partially alleviate this issue, and based on this, we construct a new dataset via a strict LLM+human filtering pipeline, which achieves higher quality and broader coverage than existing training and test sets.（3） In addition, we train a verifier with short CoT using SFT+RL, which improves verification performance without significantly increasing output length.
> > >
> > > Our novelty lies primarily in these aspects. The SFT+RL training scheme is used as a practical tool in our solution pipeline. Since existing techniques in this area already perform well, we did not further optimize this part. We believe this does not diminish the overall originality and novelty of the work.
> > >
> > > In summary, the main contribution of SCI-Verifier is to identify a critical issue in current LLM training and evaluation for scientific reasoning, and to address it with a unified model and pipeline through carefully constructed data and the training of a CoT-based verifier.
> > >
> > > **Weakness4.There is a risk that models overfit to transformation templates rather than general principles of equivalence. The authors partly mitigate this with human curation for the test set, but a strict “held‑out transformation family” evaluation would strengthen the generalization claim.**
> > >
> > > A5. First, regarding the proportion of generated data: in the test set, each domain contains 150 equivalence-augmented (generated) examples and 350 real LLM answers, so performance is still primarily driven by the verifier’s accuracy on real answers. In the training set, the vast majority of data comes from real LLM outputs, with a real:generated ratio of roughly 9:1, which is far from a regime where the model could overfit to a small set of templates.
> > >
> > > Second, concerning the effect of removing generated data, Table 4 in the main paper reports results on VerifierBench and VerifyBench-Hard, which do not explicitly include our generated templates. The performance gains of SCI-Verifier on these two datasets indicate that our training set has not caused the trained model to overfit.
> > >
> > > Third, the table below shows performance on our own test set after removing all generated data. Our model still performs well in this setting, indicating that it does not overfit to the generated templates during training.  On this subset of the test set, our SCI-Verifier also demonstrates a clear performance advantage, and the ranking of different verifiers is largely consistent with that on VerifierBench and VerifyBench. This consistency indicates that our test set construction is reasonable and that it can reliably evaluate the capability of each verifier.
> > >
> > > Table R1. Performance of different verifiers on SCI-VerifyBench after removing all generated data.
> > >
> > > |                     | Math  | Physics | Chmistry | Biology | QA    | Avg.  |
> > > | ------------------- | ----- | ------- | -------- | ------- | ----- | ----- |
> > > | GPT-5               | 89.71 | 80.28   | 86.85    | 84.57   | 82.28 | 84.74 |
> > > | Gemini-2.5-Flash    | 86.28 | 75.71   | 87.14    | 79.43   | 84.85 | 82.68 |
> > > | CompassVerifier-32B | 89.43 | 79.71   | 87.71    | 88.86   | 77.71 | 84.68 |
> > > | SCI-Verifier-4B     | 90.28 | 80.57   | 85.42    | 89.71   | 86.85 | 86.57 |
> > > | SCI-Verifier-8B     | 90.28 | 81.71   | 81.71    | 95.71   | 86.28 | 87.14 |

---

> > > > ### Author Response · Authors · 2025-11-21
> > > > **Official Comment by Authors**
> > > >
> > > > **Question3.Could specialized verifiers be run with a CoT‑enabled prompt (when feasible) to check whether the gap persists?**
> > > >
> > > > A6. Thank you for raising this point. The table below reports the Avg@4 results of our dedicated verifier with and without CoT. We observe no significant improvement after adding CoT for other specialized verifiers, likely because the verifiers have not been specifically optimized to exploit CoT signals.
> > > >
> > > > In contrast, our SCI-Verifier is designed with CoT reasoning in mind: under the CoT setting it not only outperforms existing verifier models, but also achieves this with substantially fewer reasoning tokens, which further validates the effectiveness of our approach.
> > > >
> > > > Table R2. The Avg@4 results for the Verifiers allowed to use reasoning.
> > > >
> > > > |                     | COT  | SCI-VerifyBench | VerifierBench |       | VerifyBench-Hard |       |
> > > > | ------------------- | ---- | --------------- | ------------- | ----- | ---------------- | ----- |
> > > > |                     |      | ACC             | ACC           | F1    | ACC              | F1    |
> > > > | xVerify-8B          | ×    | 75.16           | 78.03         | 75.53 | 83.20            | 79.60 |
> > > > | CompassVerifier-3B  | ×    | 76.68           | 82.39         | 83.37 | 86.60            | 84.16 |
> > > > |                     | √    | 76.92           | 82.43         | 83.65 | 87.20            | 84.56 |
> > > > | CompassVerifier-7B  | ×    | 76.56           | 85.56         | 84.83 | 87.50            | 84.13 |
> > > > |                     | √    | 76.28           | 85.75         | 84.95 | 87.20            | 84.02 |
> > > > | CompassVerifier-32B | ×    | 77.50           | 89.88         | 88.91 | 88.30            | 85.86 |
> > > > |                     | √    | 78.00           | 89.45         | 88.62 | 88.10            | 85.65 |
> > > > | SCI-Verifier-4B     | √    | 85.40           | 92.37         | 92.01 | 88.90            | 85.98 |
> > > > | SCI-Verifier-8B     | √    | 86.28           | 93.01         | 93.06 | 90.30            | 87.45 |
> > > >
> > > > **Question4.Do you have results where entire transformation families seen in training are held out in testing?**
> > > >
> > > > A7. Thank you for raising this question. First, we clarify that the equivalence-transformed samples used in the training set never appear in the test set.
> > > >
> > > > In Table 3, we report results on the full test set; in Figure 3, we report results on the subset with equivalence transformations. These experiments demonstrate the performance advantage of SCI-Verifier in scenarios involving complex equivalence transformations, which are pervasive in the scientific domain and thus highly relevant for both training and evaluation.
> > > >
> > > > In the table below, we further report results on the test set after removing all equivalence-transformed samples. Our model still performs well in this setting, indicating that it does not overfit to the generated templates during training.  On this subset of the test set, our SCI-Verifier also demonstrates a clear performance advantage, and the ranking of different verifiers is largely consistent with that on VerifierBench and VerifyBench. This consistency indicates that our test set construction is reasonable and that it can reliably evaluate the capability of each verifier.
> > > >
> > > > Table R3. Performance of different verifiers on SCI-VerifyBench after removing all generated data.
> > > >
> > > > |                     | Math  | Physics | Chmistry | Biology | QA    | Avg.  |
> > > > | ------------------- | ----- | ------- | -------- | ------- | ----- | ----- |
> > > > | GPT-5               | 89.71 | 80.28   | 86.85    | 84.57   | 82.28 | 84.74 |
> > > > | Gemini-2.5-Flash    | 86.28 | 75.71   | 87.14    | 79.43   | 84.85 | 82.68 |
> > > > | CompassVerifier-32B | 89.43 | 79.71   | 87.71    | 88.86   | 77.71 | 84.68 |
> > > > | SCI-Verifier-4B     | 90.28 | 80.57   | 85.42    | 89.71   | 86.85 | 86.57 |
> > > > | SCI-Verifier-8B     | 90.28 | 81.71   | 81.71    | 95.71   | 86.28 | 87.14 |

---

> > > > > ### Author Response · Authors · 2025-11-21
> > > > > **Official Comment by Authors**
> > > > >
> > > > > **Question5. Please report wall‑clock latency and cost/token comparisons vs GPT‑5 and CompassVerifier on SCI‑VerifyBench.**
> > > > >
> > > > > A8. Thank you for this question. Below we report the wall-clock latency and token counts of different models on SCI-VerifyBench. For open-source models, we run local inference on 2×H200 GPUs with 32 CPU cores, 32 GB RAM, and batch size 256; the measured time includes both model loading and the full inference process. For closed-source models such as GPT-5, we use the official API with 64 threads and measure end-to-end runtime.
> > > > >
> > > > > We observe that both versions of SCI-Verifier have shorter wall-clock latency than GPT-5 while achieving comparable performance. This is likely due to the additional network and service overhead of GPT-5. As a result, our verifier is better suited as a reward model in training pipelines, where fast turnaround is crucial.
> > > > >
> > > > > Table R4. The wall-clock latency and token counts of different models on SCI-VerifyBench.
> > > > >
> > > > > |                     | Accuracy | token  | wall‑clock latency (s) |
> > > > > | ------------------- | -------- | ------ | ---------------------- |
> > > > > | GPT-5               | 84.92    | 384.59 | 774.32                 |
> > > > > | CompassVerifier-3B  | 76.92    | 192.00 | 116.10                 |
> > > > > | CompassVerifier-7B  | 76.28    | 162.34 | 129.74                 |
> > > > > | CompassVerifier-32B | 78.00    | 212.04 | 332.31                 |
> > > > > | SCI-Verifier-4B     | 85.40    | 485.13 | 231.23                 |
> > > > > | SCI-Verifier-8B     | 86.28    | 490.66 | 261.92                 |
> > > > >
> > > > > **Reference**
> > > > >
> > > > > [1] IMO. International Mathematical Olympiad - Wikipedia — en.wikipedia.org. https://en.wikipedia.org/wiki/International_Mathematical_Olympiad.
> > > > >
> > > > > [2] IPhO. International Physics Olympiad - Wikipedia— en.wikipedia.org. https://en.wikipedia.org/wiki/International_Physics_Olympiad.
> > > > >
> > > > > [3] Ji, R., Wu, Y. On ends of finite-volume noncompact manifolds of nonpositive curvature. *Invent. math.* **237**, 735–777 (2024). https://doi.org/10.1007/s00222-024-01266-0

---

> ### Author Response · Authors · 2025-11-26
> **Official Comment by Authors**
>
> Dear Reviewer,
>
> Thank you once again for your thoughtful and constructive feedback. We kindly wish to confirm whether our rebuttal has satisfactorily addressed your concerns, or if there are any remaining points that might still be unclear. We would be very happy to provide any further clarification if needed.
>
> Thank you sincerely for your time and consideration.

---

> > ### Comment · Reviewer_x8GA · 2025-11-28
> >
> > Thank you for the detailed response. Well done. My concerns have been addressed, and I will raise my score accordingly. Please make sure to incorporate the promised clarifications and additional content into the revised manuscript.

---

### Official Review · Reviewer_9b7g · 2025-10-27

**Soundness:** 3
**Presentation:** 3
**Contribution:** 3
**Rating:** 4
**Confidence:** 5

**Summary:**

This paper presents SCI-VerifyBench, a cross-disciplinary benchmark covering math, physics, biology, chemistry, and general scientific QA, and SCI-Verifier, a reasoning-augmented verifier. It addresses scientific verification challenges via data (high-quality benchmark) and model (two-stage post-training) solutions, outperforming baselines and matching GPT-5.

**Strengths:**

1. SCI-VerifyBench uses real LLM responses and domain-specific equivalence transformations, ensuring diverse, high-quality data across 5 disciplines, filling gaps in existing narrow-coverage benchmarks.
2. SCI-Verifier integrates logical reasoning via SFT and RL, enabling accurate complex equivalence judgments and stable outputs, outperforming rule-based or prompt-dependent methods.
3. It shows strong cross-disciplinary performance, matches closed-source GPT-5 with 8B parameters, and maintains accuracy under prompt variations, aiding reliable LLM use in science.

**Weaknesses:**

1. I have doubts about the "w/ thinking" setup. RLVR enables large-scale RL training through a rule-based verifier, and specialized verifiers also need to minimize inference costs to prevent training from getting stuck during reward calculation. However, the paper does not quantify the inference efficiency of SCI-Verifier. It remains unclear whether the verifier in thinking mode can meet practical requirements during large-scale RL training.

2. Continuing the previous question, in the process of calculating rewards in RL, what is the role of steps other than Final Judgment, such as Validity Check and Comparison of Final Answers? Are they directly ignored? I suggest the authors conduct small-scale RL experiments to demonstrate the advantages of SCI-Verifier during the training process (refer to [1]).

3. It is recommended that the authors add more verifier cases under the thinking mode to facilitate a better understanding of the verification process of SCI-Verifier.

4. There seems to be an inversion in the citations here: "VerifyBench (Li et al., 2025) covers general, logical, mathematical reasoning, and VerifyBench (Yan et al., 2025) has 4,000 expert-level questions across STEM domains". The correct citation should be: VerifyBench (Li et al., 2025) has 4,000 expert-level questions across STEM domains, and VerifyBench (Yan et al., 2025) covers general, logical, mathematical reasoning.

[1] Huang Y, Zeng W, Zeng X, et al. Pitfalls of Rule-and Model-based Verifiers--A Case Study on Mathematical Reasoning[J]. arXiv preprint arXiv:2505.22203, 2025.

**Questions:**

Please refer to the Weaknesses.

---

> ### Author Response · Authors · 2025-11-21
> **Official Comment by Authors**
>
> **Weakness1&Weakness2. It remains unclear whether the verifier in thinking mode can meet practical requirements during large-scale RL training. What is the role of steps other than Final Judgment, such as Validity Check and Comparison of Final Answers? I suggest the authors conduct small-scale RL experiments to demonstrate the advantages of SCI-Verifier during the training process.**
>
> A1&A2. Thank you for raising these questions. We address all concerns related to incorporating SCI-Verifier into the training process in this response. First, note that the validity checks and the comparison between the model’s final answer and the reference solution are explicitly integrated into the reward computation. Concretely, if any of these checks fails, we assign a reward of 0 to the current sampled trajectory. Only when all conditions are satisfied do we consider the model’s output to match the ground-truth answer and assign a positive reward.
>
> For the experiments where we introduce SCI-Verifier into RL training, we use Qwen3-4B-Base as the base model and compare different reward models on math and physics tasks in terms of both performance and wall-clock training cost (see the table below). For the training data, on the math domain we use the open-r1/OpenR1-Math-220k dataset and randomly sample 1,000 examples as the training set; on the physics side we randomly sample 1,000 examples from desimfj/PHYSICS. Our goal here is to validate the effectiveness of the verifier, rather than to optimize the RL algorithm, so we intentionally use relatively small training sets. For evaluation, we adopt AIME24, AIME25, and MATH500 as math benchmarks, and OlympiadBench (Physics) and GPQA (Physics) as physics benchmarks, and we report Avg@32 on all of them. In all configurations, RL training is run on 4×H200 GPUs. When a LLM verifier is used, we additionally deploy the verifier as a reward model on 2×H200 GPUs. We report results after 80 training steps for every configuration to ensure a fair comparison in both performance and time.
>
> In Tables R1 and R2 (Math and Physics), we observe that using an LLM as the reward model yields better trained performance than using rule-based rewards alone. This is mainly because the LLM-based reward can correctly assess more cases where simple rules fail, at the cost of increased training time. Compared with other LLM-based reward models, SCI-Verifier achieves higher final performance, while the additional training time relative to the rule-only setting remains within an acceptable range. This demonstrates the effectiveness of SCI-Verifier for RL training and suggests its potential applicability to larger-scale settings.
>
> Table R1. The results for using different Verifiers as Reward Model for Math training.
>
> | Math                | AIME24 | AIME25 | MATH500 | Training Step | Training Time |
> | ------------------- | ------ | ------ | ------- | ------------- | ------------- |
> | Qwen3-4B-Base       | 3.54   | 2.5    | 49.8    | \             | \             |
> | Rule                | 13.64  | 6.75   | 76.0    | 80            | 8.5h          |
> | CompassVerifier-7b  | 19.48  | 13.78  | 77.45   | 80            | 9.8h          |
> | CompassVerifier-32b | 17.55  | 14.45  | 76.8    | 80            | 10.0h         |
> | Xverify-8b          | 16.78  | 15.75  | 76.34   | 80            | 8.9h          |
> | Qwen3-30b           | 18.45  | 11.34  | 78.65   | 80            | 14.0h         |
> | SCI-Verify4b        | 20.85  | 16.86  | 78.0    | 80            | 9.5h          |
> | SCI-Verify8b        | 20.75  | 17.54  | 79.35   | 80            | 9.7h          |
>
> Table R2. The results for using different Verifiers as Reward Model for Physics training.
>
> | Physics             | Olympiadbench(Physics) | GPQA(Physics) | Training Step | Training Time |
> | ------------------- | ---------------------- | ------------- | ------------- | ------------- |
> | Qwen3-4B-Base       | 11.97                  | 7.93          | \             | \             |
> | Rule                | 23.08                  | 28.19         | 80            | 8.3h          |
> | CompassVerifier-7b  | 24.22                  | 28.87         | 80            | 9.5h          |
> | CompassVerifier-32b | 24.89                  | 29.58         | 80            | 10.2h         |
> | Xverify-8b          | 23.58                  | 28.23         | 80            | 9.0h          |
> | Qwen3-30b           | 24.05                  | 29.34         | 80            | 14.9h         |
> | SCI-Verify4b        | 25.64                  | 33.04         | 80            | 9.2h          |
> | SCI-Verify8b        | 25.87                  | 33.23         | 80            | 9.4h          |

---

> > ### Author Response · Authors · 2025-11-21
> > **Official Comment by Authors**
> >
> > **Weakness3.It is recommended that the authors add more verifier cases under the thinking mode to facilitate a better understanding of the verification process of SCI-Verifier.**
> >
> > A3. Thank you for raising this issue. In the current version of the PDF, we have updated Appendix E.1 with more examples. Overall, adding CoT-based verification makes it easier to correctly judge some challenging cases, which is important for further advancing the capabilities of large models. Here we present one example; additional examples can be found in Appendix E.1.  In this setting, a verifier output of A indicates that it judges the answer to be correct, while an output of B indicates that it judges the answer to be incorrect.
> >
> > ```LaTeX
> > \textbf{Question}:
> > Let $A B C D$ be a quadrilateral with $A=(3,4), B=(9,-40), C=(-5,-12), D=(-7,24)$. Let $P$ be a point in the plane (not necessarily inside the quadrilateral). Find the minimum possible value of $A P+B P+C P+D P$.
> >
> >
> > \textbf{Gold Ans}: $16\sqrt{17}+8 \sqrt{5}$
> >
> > \textbf{LLM Ans  (extracted)}: $66+8\sqrt5$
> >
> > \textbf{Gold Judgment}: False
> >
> > \textbf{Verifier Judgment:} To evaluate whether the candidate's final answer matches the provided standard answer, we will follow the evaluation protocol step by step.
> >
> > Analysis Step by Step
> >
> > 1. Check Validity: The candidate's answer is complete and does not show any signs of repetition or refusal to answer. Therefore, we can proceed to compare the final answer directly.
> >
> > 2. Compare Candidate’s Final Answer vs. Gold Answer:
> > - The candidate's final answer is ($66 + 8\sqrt{5}$).
> > - The gold answer is ($16\sqrt{17} + 8\sqrt{5}$).
> >
> > We need to determine if these two expressions are equivalent or if there is a significant difference. Let's break down the components:
> > - Both answers contain the term ($8\sqrt{5}$), which is common.
> >
> > - The remaining terms are (66) and ($16\sqrt{17}$).
> >
> > To check for equivalence, we can compare the numerical values of these terms:
> > - (66) is a constant.
> > - ($16\sqrt{17}$) is approximately ($16 \times 4.123 = 65.968$).
> >
> > Clearly, (66) and (65.968) are very close in value, but they are not exactly the same. The difference is minimal, but it is still present.
> >
> > 3. Provide a Thorough Reasoning Chain:
> > - The candidate's answer ($66 + 8\sqrt{5}$) is very close to the gold answer ($16\sqrt{17} + 8\sqrt{5}$), but not exactly the same.
> > - The difference between (66) and ($16\sqrt{17}$) is approximately (0.032), which is negligible in most practical contexts but not zero.
> >
> > Final Judgment:
> >
> > Given that the candidate's final answer ($66 + 8\sqrt{5}$) is very close to the gold answer ($16\sqrt{17} + 8\sqrt{5}$) but not exactly the same, we can conclude that the candidate's answer is incorrect.
> >
> > Therefore, the final judgment is:[boxed\{B\}].
> > ```
> >
> > **Weakness4.There seems to be an inversion in the citations.**
> >
> > A4. Thank you for raising this issue. We have already corrected it in the current version of the PDF.

---

> > > ### Comment · Reviewer_9b7g · 2025-11-21
> > >
> > > Thank you for your detailed response.
> > >
> > > I think these results and discussions have solved my concerns so I will increase my score to 6.

---

> > > > ### Author Response · Authors · 2025-11-21
> > > > **Official Comment by Authors**
> > > >
> > > > Thank you very much for your thoughtful follow-up and for taking our response into consideration. We sincerely appreciate your time, your constructive feedback, and your decision to raise the score to 6.

---

### Official Review · Reviewer_w2ZX · 2025-10-28

**Soundness:** 3
**Presentation:** 3
**Contribution:** 2
**Rating:** 6
**Confidence:** 2

**Summary:**

The paper introduces SCI‑VerifyBench, a cross‑disciplinary benchmark for verifying the correctness of LLM answers in scientific domains (math, physics, chemistry, biology, and general QA), and SCI‑Verifier, a reasoning‑augmented verifier trained via SFT followed by RL (DAPO/GRPO‑style). The benchmark mixes real LLM responses and synthetic, domain‑specific equivalence transformations. The model generates short chain‑of‑thought before issuing a Correct/Incorrect judgment. Experiments show SCI‑Verifier‑8B performs comparably to or better than large closed models on SCI‑VerifyBench and two external benchmarks (VerifierBench, VerifyBench‑Hard), and is robust to prompt variants.

The paper focuses on an important gap: reliable verification of scientific answers. The dataset construction is well‑documented, and the training recipe is clearly summarized. However, the definitions of equivalence transformations for several tasks appear loose and subjective, potentially giving label ambiguity.

**Strengths:**

+ The motivation for reasoning in verification is clear. The authors show consistent gains when enabling CoT on two benchmarks, directly motivating a reasoning‑augmented verifier (Fig. 1 and 5).

+ The cross‑disciplinary benchmark is valuable, and the idea of using targeted equivalence transformations is interesting. The benchmark fills the breadth gap compared to previous works, covering five domains with 500 samples per domain. Concrete equivalence transformation templates are provided (Fig. 2 and Appendix A).

+ The experimental evaluations are comprehensive. The results on SCI-VerifyBench include both closed and open models with specific verifiers. Generalization to VerifierBench and VerifyBench‑Hard is also reported (Table 4). SCI‑Verifier surpasses or matches strong baselines across domains (Table 3), and excels on equivalence‑augmented tests where others degrade sharply (Fig. 3). This directly targets the paper’s core problem.

**Weaknesses:**

- There may be selection bias in benchmark construction. The test set intentionally prioritizes model‑disagreement samples for human annotation (Sec. 3.2: “retain only samples where five LLMs disagree”), which may skew distribution toward hard/atypical cases. This complicates external validity. Furthermore, “samples with disagreement between human experts and LLMs are preferentially included” in the test set (Sec. 3.2), possibly amplifying adversarial difficulty beyond natural settings. Another potential issue is the extremely long average response length (2,980 tokens; Table 2), which suggests contexts unlike many real deployments, potentially overemphasizing long‑context quirks.

- The definitions of equivalence transformation are loose or subjective in several domains. Chemistry prompts treat descriptors (high boiling point, room temperature, solvent class) as equivalent to numeric and speciﬁc answers, which is semantic similarity rather than strict equivalence. This risks label ambiguity. Biology prompts allow “conservative substitutions/synonymous sequences”, which need not preserve exact answers without explicit structural/functional constraints. QA prompts accept generic paraphrases, where scope/entailment nuances could flip correctness without careful normalization rules.

- The evaluation may lack statistical rigor. Table 3 and 4 report single accuracies without confidence intervals or multi‑seed variance, which is especially important for RL‑trained models. Prompt‑robustness Table 5 presents point estimates only; no variability across runs or bootstrap CIs. Fig. 3 (p. 7) shows sharp drops on equivalence stress tests but omits per‑domain n/CIs, hindering significance assessment.

- Some concerns:

Captions of Tables 3-4 specify: "Specialized verifiers use default prompts, while all other models are allowed to use reasoning". This design may disadvantage specialized verifiers that could also benefit from tailored prompts or concise CoT.

Average token counts differ drastically across models (e.g., many thousands for some reasoning LLMs vs 1.00 for specialized verifiers; Table 4), implying heterogeneous inference modes and costs that complicate fairness of accuracy‑only comparisons.

While Sec. 5.1 references Appendix A.2 for prompt details, it is unclear that each baseline was evaluated under its best prompt setting, especially for specialized verifiers. No direct evidence found in the manuscript.

The alignment reward (R_{align,i}) (Eq. 5) is not defined operationally.

The policy ratio (r_{i,t}(\theta)) in Eq. 3 is never explicitly defined.

The over‑long penalty uses (-\infty) for responses exceeding (L_{\max}+L_{buffer}) (Eq. 6), which is not implementable as‑is; practical handling (e.g., very large negative constants) is unspecified.

**Questions:**

See Weakness.

---

> ### Author Response · Authors · 2025-11-21
> **Official Comment by Authors**
>
> **Weakness1. There may be selection bias in benchmark construction. 1.The current selection is biased toward cases where models disagree with each other and also differ from human judgments. 2.The 2,980-token average response length (Table 2) is atypical of real deployments and may overemphasize long-context quirks.**
>
> A1. Thank you for your comments on our benchmark construction. Your concern has two parts, which we address separately.
>
> A1.1. First, regarding the disagreement between LLMs and between LLMs and human experts: we agree this makes the dataset more challenging. This is intentional. Our goal is to evaluate verifier accuracy under difficult conditions, which is crucial for complex scientific domains and for training/evaluating LLMs targeting top-tier tasks in math/physics and other sciences. In such settings, answers are much more complex in form (e.g., IMO-level solutions vs. GSM8K), so verifiers must be tested in correspondingly complex environments. Simpler settings are already covered by benchmarks such as VerifyBench and VerifierBench, on which we also report results in Table 4 of the original manuscript.
>
> Second, we apologize if our description was unclear. Actually, in the training set, a large portion of labels come from cases where multiple LLMs fully agree. Given the scale of the training data, full manual labeling is infeasible, and using consensus among LLMs is how we ensure annotation quality at scale. Therefore, the training data does not exhibit the kind of bias implied by the concern.  The trained model (SCI-Verifier) performs well on both simple benchmarks (e.g., VerifierBench) and more complex benchmarks (SCI-VerifyBench), providing evidence that it does not overfit to these hard examples.  For these reasons, we believe our dataset construction is reasonable and well-suited to improving and evaluating verifier capability in challenging regimes.
>
> A1.2. Regarding response length: this refers to the length of the full LLM reasoning trace, not the final answer. We instruct the LLM to place the final answer inside `\boxed{}` and only this boxed segment is fed to the verifier. Thus, the effective average input length for the verifier is about 24.98 tokens rather than 2980.43, so there is no issue of overly long context for the verifier.
>
> **Weakness2. The definitions of equivalence transformation are loose or subjective in several domains.**
>
> A2. We thank the reviewer for this insightful question and agree that introducing equivalence notions inevitably involves some degree of approximation. However, such equivalence is pervasive in how correctness is judged across essentially all scientific disciplines: for example, trigonometric identities in mathematics, tolerable decimal rounding in physics, or equivalent molecular formulas in chemistry and biology. These forms of equivalence are routinely accepted not only in exam grading[1,2], but also in state-of-the-art scientific derivations and published results[3].
>
> To control the approximation introduced by equivalence in our benchmarks, we proceed differently for evaluation and training. For the dev/test sets, multiple human experts label each example with access to both the question and the problem requirements, and any approximation that does *not* meet the stated requirements (e.g., an overly large rounding error) is explicitly marked as incorrect. Our evaluation strictly follows these human labels, which effectively filters out unacceptable approximate equivalences.
>
> For the training set, fully manual annotation is infeasible due to scale. Instead, we retain only those examples on which multiple LLMs agree and use this consensus as training supervision. This inevitably introduces a small amount of approximate-equivalence noise (cases that a strict human grader might reject), but in practice this behaves as reasonable training noise rather than harmful bias. Empirically, our model’s leading performance on both VerifierBench and VerifyBench-hard in Table 4 demonstrates that this design does not degrade, and in fact supports, robust generalization. Moreover, in a human audit of a sampled subset of the training data, we found that when LLMs reach consensus, the label accuracy exceeds 95%, indicating that the residual noise level is well within an acceptable range.
>
> In summary, incorporating equivalence-based data does not cause severe overfitting or undermine rigor; on the contrary, it is necessary for a verifier operating in scientific domains, where well-controlled forms of equivalence are an integral part of how correct reasoning and answers are defined.

---

> > ### Author Response · Authors · 2025-11-21
> > **Official Comment by Authors**
> >
> > **Weakness3.The evaluation may lack statistical rigor. Table 3 and 4 report single accuracies without confidence intervals or multi‑seed variance, which is especially important for RL‑trained models.**
> >
> > A3. Thank you for this question. In the table below, we report the Avg@4 results for the models corresponding to Tables 3 and 4.  We also provide Avg@4 and the variance for the results in Table 5. Since each benchmark contains more than 1,500 examples, the variance of these test results is small.
> >
> > Table R1. The Avg@4 results for the verifiers on VerifierBench and VerifyBench-Hard (corresponding to Tables 3 and 4)
> >
> > |                             | SCI-VerifyBench | VerifierBench |            | VerifyBench-Hard |            |
> > | --------------------------- | --------------- | ------------- | ---------- | ---------------- | ---------- |
> > |                             | acc             | acc           | f1         | acc              | f1         |
> > | Qwen2.5-72B-Instruct        | 75.00±0.14      | 82.61±0.25    | 81.67±0.22 | 85.20±0.12       | 81.31±0.21 |
> > | Qwen3-30B-A3B-Instruct-2507 | 78.16±0.26      | 88.78±0.10    | 88.88±0.15 | 88.70±0.11       | 85.03±0.30 |
> > | LLaMa-3.3-70B-Instruct      | 78.64±0.25      | 79.84±0.11    | 79.00±0.13 | 85.20±0.22       | 81.10±0.20 |
> > | Qwen3-8b                    | 74.52±0.31      | 85.55±0.23    | 85.56±0.22 | 84.40±0.12       | 79.58±0.11 |
> > | Qwen3-30B-A3B-Thinking-2507 | 76.96±0.21      | 90.42±0.14    | 90.05±0.15 | 88.60±0.21       | 84.92±0.23 |
> > | xVerify-8B                  | 75.16±0.20      | 78.03±0.18    | 75.53±0.14 | 83.20±0.12       | 79.60±0.14 |
> > | CompassVerifier-3B          | 76.92±0.22      | 82.39±0.12    | 83.37±0.21 | 86.60±0.23       | 84.16±0.18 |
> > | CompassVerifier-7B          | 76.28±0.14      | 85.56±0.16    | 84.83±0.23 | 87.50±0.21       | 84.13±0.16 |
> > | CompassVerifier-32B         | 78.23±0.21      | 89.65±0.06    | 88.91±0.08 | 88.30±0.25       | 85.86±0.26 |
> > | SCI-Verifier-4B             | 85.40±0.12      | 92.37±0.11    | 92.01±0.15 | 88.90±0.12       | 85.98±0.10 |
> > | SCI-Verifier-8B             | 86.28±0.08      | 93.01±0.10    | 93.06±0.20 | 90.30±0.18       | 87.45±0.22 |
> >
> > Table R2. The Avg@4 results for the comparison of model robustness across different prompts. our: default prompt; other: modified prompt (Corresponding to Table5).
> >
> > |                             | SCI-VerifyBench |            | VerifyBench-Hard |            |
> > | --------------------------- | --------------- | ---------- | ---------------- | ---------- |
> > |                             | our             | other      | our              | other      |
> > | Qwen3-30B-A3B-Instruct-2507 | 78.16±0.26      | 76.72±0.22 | 88.70±0.11       | 75.40±0.15 |
> > | GPT-oss-20b                 | 70.32±0.45      | 78.08±0.54 | 85.90±0.52       | 79.50±0.55 |
> > | CompassVerifier-3B          | 76.92±0.22      | 74.52±0.25 | 86.60±0.23       | 79.30±0.26 |
> > | CompassVerifier-7B          | 76.28±0.14      | 76.44±0.20 | 87.50±0.21       | 80.30±0.22 |
> > | CompassVerifier-32B         | 78.23±0.21      | 81.00±0.15 | 88.30±0.25       | 84.70±0.18 |
> > | SCI-Verifier-4B             | 85.40±0.12      | 84.90±0.15 | 88.90±0.12       | 88.30±0.22 |
> > | SCI-Verifier-8B             | 86.28±0.08      | 85.50±0.12 | 90.30±0.18       | 89.70±0.25 |
> >
> > **Concern 1. Specialized verifiers should also be tested for potential gains from adding CoT.**
> >
> > A4. Thank you for raising this point. The table below reports the Avg@4 results of the specialized verifiers with and without CoT. For other verifier baselines, we observe that they are either not applicable to CoT-style generators (e.g., XVerify), or they show essentially no performance improvement when CoT is enabled (e.g., CompassVerifier), largely because they are not explicitly optimized for CoT inputs. In contrast, our SCI-Verifier is designed with CoT reasoning in mind: under the CoT setting it not only outperforms existing verifier models, but also achieves this with substantially fewer reasoning tokens, which further validates the effectiveness of our approach.
> >
> > Table R3. The Avg@4 results for the Verifiers allowed to use reasoning (Corresponding to Concern1).
> >
> > |                     | COT  | SCI-VerifyBench | VerifierBench |       | VerifyBench-Hard |       |
> > | ------------------- | ---- | --------------- | ------------- | ----- | ---------------- | ----- |
> > |  | | ACC| ACC  | F1    | ACC  | F1    |
> > | xVerify-8B  | ×    | 75.16  | 78.03 | 75.53 | 83.20  | 79.60 |
> > | CompassVerifier-3B  | ×    | 76.68 | 82.39  | 83.37 | 86.60  | 84.16 |
> > |  | √    | 76.92  | 82.43 | 83.65 | 87.20 | 84.56 |
> > | CompassVerifier-7B  | ×    | 76.56 | 85.56 | 84.83 | 87.50 | 84.13 |
> > |  | √    | 76.28 | 85.75| 84.95 | 87.20 | 84.02 |
> > | CompassVerifier-32B | ×    | 77.50  | 89.88| 88.91 | 88.30 | 85.86 |
> > |     | √    | 78.00 | 89.45  | 88.62 | 88.10| 85.65 |
> > | SCI-Verifier-4B     | √    | 85.40 | 92.37| 92.01 | 88.90| 85.98 |
> > | SCI-Verifier-8B     | √    | 86.28 | 93.01 | 93.06 | 90.30 | 87.45 |

---

> > > ### Author Response · Authors · 2025-11-21
> > > **Official Comment by Authors**
> > >
> > > **Concern 2. Average token counts differ drastically across models.**
> > >
> > > A5. Thank you for this valuable question. We fully agree with your observation. Our comparison between long chain of thought outputs produced by general reasoning models and the much shorter outputs of specialized verifiers, which nevertheless reach similar accuracy, is intended to show that current dedicated verifiers do not raise the upper bound of verification accuracy; they only match the performance of reasoning models while producing shorter reasoning traces. As shown in Table R3 in our response to Concern 1, adding chain of thought inputs to existing verifier baselines also does not lead to a noticeable improvement in accuracy. This situation is unsatisfactory for applications that require a higher ceiling on verification reliability.
> > >
> > > In contrast, our proposed SCI Verifier significantly improves verification accuracy while using relatively short chain of thought, requiring fewer than 500 tokens on SCI-VerifyBench, whereas general reasoning models typically need more than 1000 tokens. This combination of higher accuracy and compact reasoning traces is crucial for efficient training and evaluation in complex verification scenarios.
> > >
> > > **Concern 3. It is unclear that each baseline was evaluated under its best prompt setting.**
> > >
> > > A6. Yes. For each baseline, we report results under its best-performing configuration, following the best settings reported in the original paper.
> > >
> > > **Concern 4. The alignment reward (R_{align,i}) (Eq. 5) is not defined operationally.**
> > >
> > > A7. Thank you for pointing out that we omitted this definition. R_{\text{align}, i} denotes whether the final prediction for the i-th example matches the ground-truth answer, and takes a value of 0 or 1. We have already updated this in the revised manuscript.
> > >
> > > **Concern 5. The policy ratio (r_{i,t}(\theta)) in Eq. 3 is never explicitly defined.**
> > >
> > > A8. Thank you for pointing out that we omitted this definition.
> > > $$
> > > r_{i,t}(\theta)
> > >
> > > = \frac{\pi_{\theta}\bigl(o_{i,t} \mid q, o_{i,<t}\bigr)}
> > >   {\pi_{\theta_{\text{old}}}\bigl(o_{i,t} \mid q, o_{i,<t}\bigr)}
> > > $$
> > > Here, $\pi_{\theta}$ and $\pi_{\theta old}$ denote the updated policy and the pre-update policy, respectively, which is consistent with the formulation used in GRPO. We have already updated this in the revised manuscript.
> > >
> > > **Concern 6.The over‑long penalty.**
> > >
> > > Q9. Thank you for pointing out that we omitted this definition.  In practice, when the answer length exceeds the upper limit, we set the penalty in Eq. 6  to $\lambda_{penalty}$, which is typically chosen between 1.0 and 1.2. We have already updated this in the revised manuscript.
> > >
> > > **Reference**
> > >
> > > [1] IMO. International Mathematical Olympiad - Wikipedia — en.wikipedia.org. https://en.wikipedia.org/wiki/International_Mathematical_Olympiad.
> > >
> > > [2] IPhO. International Physics Olympiad - Wikipedia— en.wikipedia.org. https://en.wikipedia.org/wiki/International_Physics_Olympiad.
> > >
> > > [3] Ji, R., Wu, Y. On ends of finite-volume noncompact manifolds of nonpositive curvature. *Invent. math.* **237**, 735–777 (2024). https://doi.org/10.1007/s00222-024-01266-0

---

> ### Author Response · Authors · 2025-11-26
> **Official Comment by Authors**
>
> Dear Reviewer,
>
> Thank you once again for your thoughtful and constructive feedback. We kindly wish to confirm whether our rebuttal has satisfactorily addressed your concerns, or if there are any remaining points that might still be unclear. We would be very happy to provide any further clarification if needed.
>
> Thank you sincerely for your time and consideration.

---

### Author Response · Authors · 2025-12-03
**Official Comment by Authors**

Thank you very much to all reviewers, as well as the ACs and PCs, for your time and effort.  We sincerely appreciate the reviewers’ positive recognition below.

-  **Clear motivation and problem framing for scientific verification and reasoning.** (Reviewer w2ZX, x8GA, Hp78)
- **Broad, high-quality cross-disciplinary benchmark with realistic equivalence transformations.** (Reviewer w2ZX, 9b7g, x8GA, Hp78)
- **Well-designed reasoning-augmented verifier combining CoT.** (Reviewer 9b7g, x8GA)
- **Strong and robust performance across scientific domains and benchmarks.** (Reviewer w2ZX, 9b7g, x8GA)
- **Thorough and convincing experimental evaluation.** (Reviewer w2ZX, x8GA,Hp78)
- **Clear writing and comprehensive end-to-end execution (data, benchmark, verifier).** (Reviewer Hp78)

We are pleased that, after considering both the original submission and our rebuttal, **all reviewers indicated that their concerns have been effectively addressed and have provided positive overall assessment**. Below we briefly summarize the main shared concerns and how our revision responds to them:

- **Data selection bias.** For the test set, we mitigate potential bias through a two-stage filtering procedure that combines LLM-based pre-screening with careful human expert verification, ensuring that selection bias is negligible. For the training set, we mainly retain examples on which multiple LLMs reach consensus, which further reduces bias in the supervision signal. (Please refer to Reviewer w2ZX: Weakness 1, Concern 2; Reviewer x8GA: Weakness 1, Question 1, Weakness 4, Question 4).
- **Definition and use of equivalence.** Equivalence transformations are essential for correctness assessment in scientific domains. We therefore enforce strict equivalence in the test set via human expert filtering, and only include a small proportion of constructed equivalence cases in the training set. The experimental results in A5 for Reviewer x8GA, which show that SCI-Verifier still performs strongly on SCI-VerifyBench even after removing all generated data（GPT-5 vs SCI-Verifier-4b vs SCI-Verifier-8b：84.74 vs 86.57 vs 87.14）, indicate that SCI-Verifier does not overfit to these equivalence patterns, supporting the soundness of our design. (Please refer to Reviewer w2ZX: Weakness 2; Reviewer x8GA: Weakness 2 & Question 2)
- **Using SCI-Verifier as a reward model in RL training.** On both math and physics data, we conduct RL training using SCI-Verifier as the reward model. Compared with purely rule-based rewards, our approach yields substantially better performance (**Math:** AIME24: 13.64 vs 20.75; AIME25: 6.75 vs 17.54; MATH500: 76.0 vs 78.35. **Physics:** Olympiadbench(Physics): 23.08 vs 25.87; GPQA(physics): 28.19 vs 33.23) while keeping the additional training time within a reasonable range (around 9.5h for SCI-Verifier and 8.5h for Rule-based) (Detailed results refer to A1&A2 for Reviewer 9b7g). (Please refer to Reviewer 9b7g: Weakness 1 & 2)
- **Case studies.** We have added detailed case analyses, including representative reasoning traces of SCI-Verifier (see our response to Reviewer 9b7g, Weakness 3, and Appendix E.1 in the revised manuscript) and error analyses that illustrate remaining failure modes and their causes (see our response to Reviewer Hp78, Weakness 1 & 2, and Appendix E.1). (Please refer to Reviewer 9b7g: Weakness 3; Reviewer Hp78: Weakness 1 & 2)
- **Novelty and contribution.** As emphasized in the revised version, the main contribution of SCI-Verifier is to identify a critical, previously under-explored issue in current LLM training and evaluation for scientific reasoning. It then addresses this issue with a unified verifier model and pipeline. This is achieved through carefully constructed data and the training of a CoT-based verifier specifically tailored to scientific domains. (Please refer to Reviewer x8GA: Weakness 3)
- **Other specialized verifiers under the CoT setting.** We further verify that, since existing verifiers are not specifically optimized for CoT, simply enabling CoT-style outputs does not yield any substantial performance improvement (see our response to Reviewer w2ZX, Concern 1, Table R3). (Please refer to Reviewer w2ZX: Concern 1; Reviewer x8GA: Question 3)

We hope these clarifications make our contributions clearer, and we are grateful again for the reviewers’ constructive feedback and positive reassessment.

---

### Meta-Review · Area_Chair_Ty6W · 2026-01-04

**Summary:**

The paper is about a cross-disciplinary (Math, Phy, Chem, etc) verification benchmark built from real LLM responses, and domain-specific equivalence transformations. Plus, the paper also contributes a reasoning augmented verifier trained with SFT+RL for scientific verification. The results outperform baselines and match GPT-5!

The reviewers gave it a 4, 4, 6 and 8. Honestly, 8 was an uninformative review, but 4,4,6 raise interesting questions -- both high level (like novelty) and many detail and clarification questions. The authors, in their part, went above and beyond and answered each point in good detail. Both reviewers who gave a 4 promised to increase scores -- one to 6 and other to some number that was eventually reverted due to ICLR policies this time.

My personal read is that most questions have been satisfactorily answered. The question of novelty still lingers, but it lingers on many papers as new ML techniques are not always contributed by a paper. Overall, I propose a weak accept.

**Reviewer Concerns:**

Many concerns. The concerns that were not adequately answered:

- novelty of methodology: SFT+RL is not built upon. Primary contribution is in dataset generation style. That said, these days ML novelty is harder to achieve. The paper makes a valuable contribution... but this may be a good reason to reject, if the number of papers are getting higher at the conference.

- aren't equivalences approximate: two reviewers raise this. Authors acknowledge but say that the proof is in the pudding, i.e., there are evaluations that show that it helps. I am comfortable at this argument.

Many other questions like why focus on large contexts, why focus on test set that is constructed based on LLM-disagreement, timing analysis, what happens if verifiers are run with coT, etc etc. In my assessment, those questions have been well answered.

**Reviewer Scores:**

6 should ideally go to 7. 8 would remain 8.
One 4 will go to 6 (since they said so).
Other 4 should also go to 6 even though their novelty question was not satisfactorily answered.

Overall, I'd treat this paper as 6,6,7,8.

---

### Decision · Program_Chairs · 2026-01-26

Accept (Poster)